# Defective metabolic programming impairs early neuronal morphogenesis in neural cultures and an organoid model of Leigh syndrome

Gizem Inak[1,2], Agnieszka Rybak-Wolf[3], Pawel Lisowski[1,2,4], Tancredi M. Pentimalli[3], René Jüttner[1], Petar Glažar[3], Karan Uppal[5], Emanuela Bottani [6], Dario Brunetti [7,8], Christopher Secker [1,9], Annika Zink [1,2,10], David Meierhofer [11], Marie-Thérèse Henke [1,12], Monishita Dey [1], Ummi Ciptasari [1], Barbara Mlody[1], Tobias Hahn[1], Maria Berruezo-Llacuna[1], Nikos Karaiskos[3], Michela Di Virgilio [1], Johannes A. Mayr [13], Saskia B. Wortmann[13,14], Josef Priller [10,15,16], Michael Gotthardt [1], Dean P. Jones[5], Ertan Mayatepek[2], Werner Stenzel[17], Sebastian Diecke [1,18], Ralf Kühn [1], Erich E. Wanker [1], Nikolaus Rajewsky [3✉], Markus Schuelke [12,19✉] & Alessandro Prigione [1,2✉]

Leigh syndrome (LS) is a severe manifestation of mitochondrial disease in children and is currently incurable. The lack of effective models hampers our understanding of the mechanisms underlying the neuronal pathology of LS. Using patient-derived induced pluripotent stem cells and CRISPR/Cas9 engineering, we developed a human model of LS caused by mutations in the complex IV assembly gene *SURF1*. Single-cell RNA-sequencing and multi-omics analysis revealed compromised neuronal morphogenesis in mutant neural cultures and brain organoids. The defects emerged at the level of neural progenitor cells (NPCs), which retained a glycolytic proliferative state that failed to instruct neuronal morphogenesis. LS NPCs carrying mutations in the complex I gene *NDUFS4* recapitulated morphogenesis defects. *SURF1* gene augmentation and PGC1A induction via bezafibrate treatment supported the metabolic programming of LS NPCs, leading to restored neuronal morphogenesis. Our findings provide mechanistic insights and suggest potential interventional strategies for a rare mitochondrial disease.

[1] Max Delbrück Center for Molecular Medicine (MDC), Berlin, Germany. [2] Department of General Pediatrics, Neonatology and Pediatric Cardiology, Duesseldorf University Hospital, Medical Faculty, Heinrich Heine University, Duesseldorf, Germany. [3] Berlin Institute for Medical Systems Biology (BIMSB), Max Delbrück Center for Molecular Medicine (MDC), Hannoversche Str 28, 10115, Berlin, Germany. [4] Institute of Genetics and Animal Biotechnology, Polish Academy of Sciences, Jastrzebiec, n/Warsaw, Magdalenka, Poland. [5] Emory University, Atlanta, GA, USA. [6] Department of Diagnostics and Public Health, University of Verona, Verona, Italy. [7] Mitochondrial Medicine Laboratory, Department of Medical Biotechnology and Translational Medicine, University of Milan, Milan, Italy. [8] Unit of Medical Genetics and Neurogenetics Fondazione IRCCS Istituto Neurologico "Carlo Besta", Milan, Italy. [9] Charité - Universitätsmedizin Berlin, Department of Neurology, Berlin, Germany. [10] Charité - Universitätsmedizin Berlin, Department of Neuropsychiatry, Berlin, Germany. [11] Max Planck Institute for Molecular Genetics, Berlin, Germany. [12] Charité - Universitätsmedizin Berlin, Department of Neuropediatrics, Berlin, Germany. [13] University Children's Hospital, Paracelsus Medical University (PMU), Salzburg, Austria. [14] Radboud Center for Mitochondrial Medicine, Department of Pediatrics, Amalia Children's Hospital, Radboudumc, Nijmegen, The Netherlands. [15] University of Edinburgh and UK DRI, Edinburgh, UK. [16] Department of Psychiatry and Psychotherapy, Klinikum rechts der Isar, Technical University Munich, Munich, Germany. [17] Charité – Universitätsmedizin, Department of Neuropathology, Berlin, Germany. [18] Berlin Institute of Health (BIH), Berlin, Germany. [19] NeuroCure Clinical Research Center, Berlin, Germany. ✉email: rajewsky@mdc-berlin.de; markus.schuelke@charite.de; alessandro.prigione@hhu.de

Mitochondrial disease represents the largest class of inborn errors of metabolism mainly comprising monogenic disorders that disrupt oxidative phosphorylation (OXPHOS)[1]. The most severe pediatric manifestation of mitochondrial disease is Leigh syndrome (OMIM #256000)[2]. Leigh syndrome (LS) affects 1 in 36,000 newborns[3] and causes lactic acidosis and symmetric lesions in the central nervous system (CNS), predominantly of basal ganglia and brainstem, leading to intellectual disability and muscle weakness with a peak of mortality before three years of age[4]. Mutations in more than 75 genes of nuclear or mitochondrial DNA (mtDNA) can cause LS[5]. The most commonly affected mitochondrial complexes in LS are complex I (CI) and complex IV (CIV)[6].

One of the most frequently mutated nuclear genes in LS is the *SURF1* gene (Surfeit locus protein 1, NM_003172.2)[7–9]. *SURF1* gene contains 9 exons, is located on chromosome 9q34, and encodes a protein that is highly conserved among eukaryotes and prokaryotes[10]. SURF1 protein is located in the mitochondrial inner membrane and is involved in the assembly of CIV, i.e., the cytochrome c oxidase (COX) (Fig. 1a). SURF1 appears particularly important for the assembly of mtDNA-encoded proteins MT-CO1, MT-CO2, and MT-CO3, which constitute the catalytic core of COX[11,12]. When the stability of this active COX site is impaired, the whole COX assembly can be compromised[13,14]. Accordingly, *SURF1* mutations lead to defective COX assembly[15,16], the absence of *SURF1* or its yeast homolog *Shy1* cause severe COX deficiency[8,17], and fibroblasts and muscle fibers carrying *SURF1* mutations display decreased COX activity[7,8,16].

There are no FDA-approved drugs to treat mitochondrial disease, and the pathophysiological mechanisms underlying the CNS defects are poorly understood[18,19]. One major obstacle for treatment development is the lack of effective model systems recapitulating the human disease course[18,19]. This is particularly the case for *SURF1* mutations[20]. *Surf1* knock-out mice failed to develop LS-like neurological phenotypes and showed prolonged lifespan[21,22]. CNS-specific knock-down of *Surf1* in flies caused COX deficiency but no neurological alterations[23]. *Surf1* knock-down in zebrafish led to developmental defects mainly in endodermal tissue and peripheral nervous system[24]. *Surf1* knock-out in piglets resulted in a severe lethal phenotype with CNS developmental delay without overt neurodegeneration and no apparent COX deficiency[25]. These species-specific differences underscore the possibility that human cells are more dependent on SURF1 function for COX assembly than those of other species. In fact, key differences in COX expression and assembly have been observed in fibroblasts from SURF1 patients compared to fibroblasts from *Surf1* knock-out mice[26,27].

Recent studies generated induced pluripotent stem cells (iPSCs) from LS patients carrying mtDNA mutations in the mitochondrial complex V (CV) gene *MT-ATP6* or the CI gene *MT-ND5*. LS patient-derived neural cells showed defective bioenergetics[28,29], decreased protein synthesis[30], impaired mitochondrial calcium homeostasis[29,31], and abnormal corticogenesis[32]. However, the molecular pathophysiology of LS caused by *SURF1* mutations remains to be elucidated.

We applied patient-specific iPSCs and CRISPR/Cas9 technology to investigate the mechanisms underlying the neuronal pathology caused by *SURF1* mutations. Single-cell transcriptomics and multi-omics analysis of mutant two-dimensional (2D) neural cultures and three-dimensional (3D) cerebral organoids pointed toward a defect of early neuronal morphogenesis. Mutant neural progenitor cells (NPCs) were unable to shift toward OXPHOS and retained a proliferative glycolytic state that failed to instruct morphogenesis. Morphogenesis defects were recapitulated in neural cells carrying mutations in the nuclear CI gene *NDUFS4* (NADH dehydrogenase [ubiquinone] iron-sulfur protein 4, NM_002495.2),

another frequent cause of LS[6,33,34] that has not been studied so far using iPSCs.

Interventions that boosted the metabolic shift of NPCs, such as lentivirus-mediated and adeno-associated virus (AAV)-mediated SURF1 gene augmentation and stimulation of mitochondrial biogenesis via bezafibrate treatment or PGC1A overexpression, restored early neuronal morphogenesis. Our findings suggest that dysfunctional neuronal wiring could be at the basis of the neurodevelopmental defects seen in LS, and indicate potential intervention strategies for this incurable mitochondrial disease in children.

## Results

**Generation of an iPSC-based model of LS caused by *SURF1* mutations.** We obtained skin fibroblasts from two LS patients belonging to two distinct consanguineous families carrying homozygous mutations in *SURF1* (Fig. 1a, b). Patient S1 harbored a c.530T>G p.(V177G) mutation (Supplementary Fig. 1a-b), while patient S2 carried a c.769G>A p.(G257R) mutation (Supplementary Fig. 1c-d). Among the two mutations, c.769G>A led to a more clinically severe phenotype and was reported to be the most common variant observed in LS patients in the Turkish community[35] to which both patient families belonged.

We generated SURF1 iPSCs (S1 and S2) with Sendai viruses (Supplementary Fig. 1e–i). We used clustered regularly interspaced short palindromic repeats (CRISPR) and high-fidelity Cas9 (eCas9), which has limited off-target effects[36], to correct the more severe mutation c.769G>A (p.G257R) on both alleles of S2 iPSCs. We generated isogenic iPSC lines that did not carry the mutation within the patient's genetic background (SURF1_NoMut) (Fig. 1c). We generated three isogenic SURF1_NoMut lines: S2_Corr1, S2_Corr2, and S2_Corr3 (Supplementary Fig. 2c-d). We increased the efficiency of knock-in events by promoting homologous direct repair (HDR) and inhibiting non-homologous end joining by co-transfecting plasmids expressing RAD52 and dominant-negative p53-binding protein 1 (dn53BP1)[37] (Supplementary Fig. 2a). In the HDR template, we introduced two silent mutations that we could trace to confirm template integration (Fig. 1c, Supplementary Fig. 2b). Using the same guide RNAs (gRNAs) and approach, we further targeted the more severe c.769G>A mutation. We introduced the c.769G>A mutation into both alleles of control (CTL) iPSCs (line C1) to generate iPSC lines that carried the mutation in a control healthy background (CTL_Mut) (Fig. 1c, Supplementary Fig. 2b). We obtained two isogenic CTL_Mut lines: C1_Mut1 and C1_Mut2.

We confirmed that genetically engineered iPSC lines (SURF1_NoMut and CTL_Mut) maintained their pluripotent identity (Supplementary Fig. 2e) and retained a normal karyotype (Supplementary Fig. 2f-g). We performed whole-genome sequencing (WGS) of S2 iPSCs and S2_Corr1 iPSCs. We used the CRISPR Gold algorithm to predict off-target effects and then interrogated the WGS datasets accordingly. We confirmed the absence of unintended events (top 58 predicted off-target sites) in S2_Corr1 compared to S2, demonstrating the reliability of our genome editing approach (Supplementary Data 1).

Altogether, we generated two sets of complementary isogenic iPSC lines. The first isogenic set carried the *SURF1* mutation c.769G>A either in a healthy genetic background (CTL_Mut) or in a patient-specific genetic background (SURF1_Mut). The second isogenic set was mutation-free either in a healthy genetic background (CTL_NoMut) or in a patient-specific genetic background (SURF1_NoMut). This genome engineering approach would enable us to dissect the specific impact of the *SURF1* mutation on the disease pathogenesis irrespective of any potential effects due to the nuclear or mitochondrial genomic

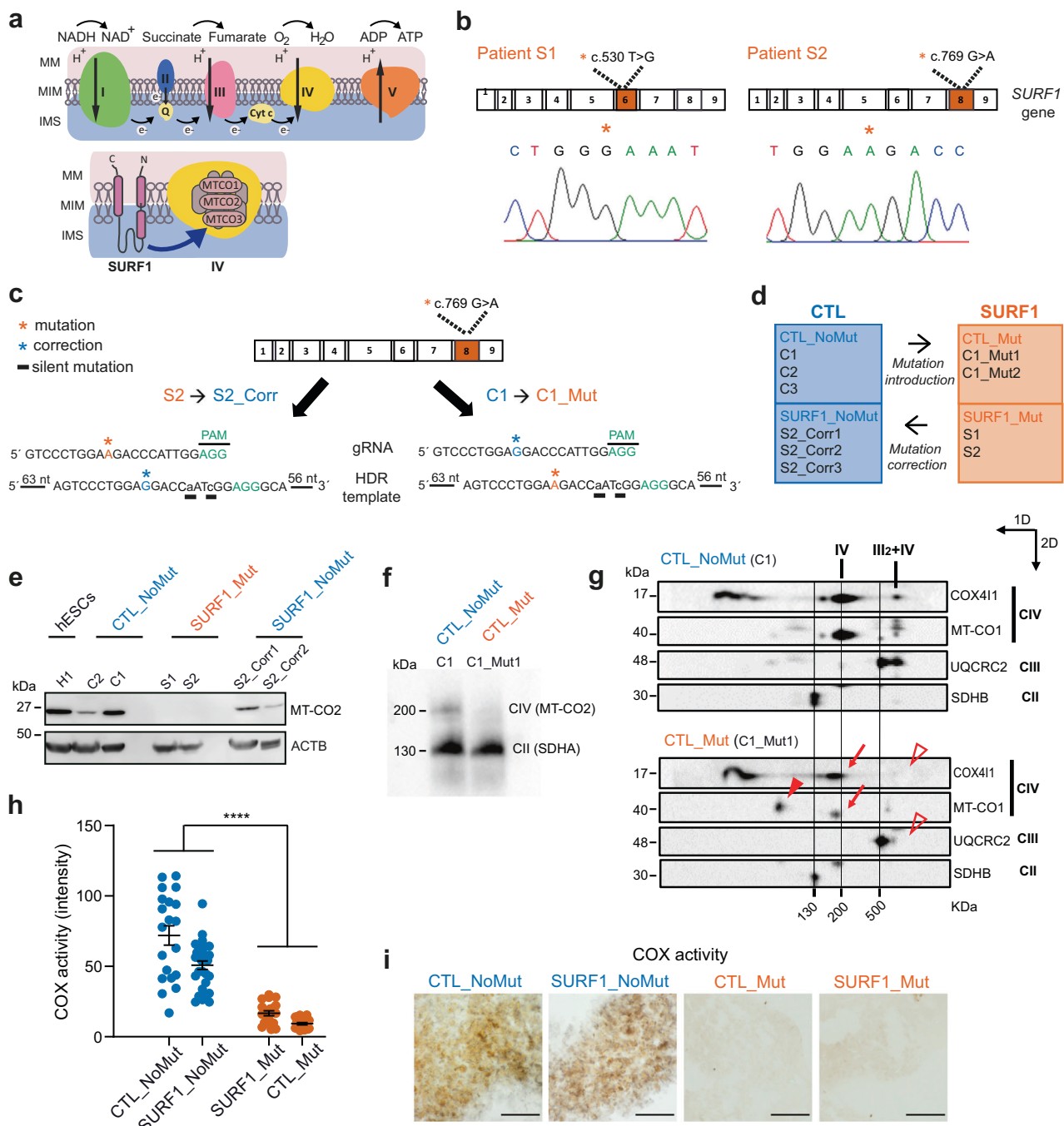

**Fig. 1 Generation of an iPSC-based model of LS due to _SURF1_ mutations. a** Mitochondrial respiratory chain and putative SURF1 function in CIV assembly. MM mitochondrial membrane, IMM inner mitochondrial membrane, IMS intermembrane space. **b** Electropherograms showing 5′ > 3′ sequences of _SURF1_ in iPSCs from patients S1 and S2 carrying the mutations c.530T > G p.(V177G) and c.769G > A p.(G257R), respectively. Orange stars: mutation site. **c** Genome editing approach to correct mutation c.769G > A in patient line S2, and to introduce the same mutation in control line C1. Orange stars: mutation site; blue stars: correction site; black underlines: artificially introduced silent mutations; gRNA: guide RNA; HDR: homology direct repair. **d** Overview of iPSC lines used in this study. **e** Immunoblot of MT-CO2 (26 kDa) in NPCs derived from hESCs (H1), CTL (CTL_NoMut: C2, C1; SURF1_NoMut: S2_Corr1, S2_Corr2), and SURF1 (SURF1_Mut: S1, S2) (_n_ = 2 independent experiments). **f** One dimensional (1D) blue-native gel electrophoresis (BNGE) analysis of mitochondrial complexes showing lack of incorporation of MT-CO2 into CIV in NPCs from CTL (CTL_NoMut: C1) and SURF1 (CTL_Mut: C1_Mut1). The structure of the complex II (CII) (visualized by SDHA antibody) was not affected (_n_ = 2 independent experiments). **g** Two-dimensional (2D) BNGE showing mitochondrial complex assembly in NPCs from CTL (CTL_NoMut: C1) and SURF1 (CTL_Mut: C1_Mut1). Red arrows: structurally impaired CIV migrating at lower molecular weight (detected by MT-CO1 and COX4I1 antibodies); red arrowhead: accumulation of COX assembly intermediates (detected by MT-CO1 antibody); open red arrowheads: loss of III2 + IV supercomplex (detected by COX4I1 and UQCRC2 antibodies). We used CIII (detected by UQCRC2 antibody) and CII (detected by SDHB antibody) to align CTL blot with SURF1 blot (_n_ = 2 independent experiments). **h, i** Quantification and representative images of COX activity intensity in CTL NPCs (CTL_NoMut: C1, C2, C3; SURF1_NoMut: S2_Corr1, S2_Corr2) and SURF1 NPCs (SURF1_Mut: S1, S2; CTL_Mut: C1_Mut1) (mean ± s.e.m.; each dot represents a biological replicate; _n_ = 10 biological replicates per line over three independent experiments; ****_p_ < 0.0001 CTL vs. SURF1; two-sided Mann–Whitney U test). Scale bar: 500 nm.

backgrounds, whose interplay has been suggested to contribute to the development of mitochondrial disorders[38].

We grouped our samples into "controls" (CTL) and "diseased" (SURF1). The CTL group included (i) CTL_NoMut lines that we derived from healthy individuals (C1, C2, C3) and (ii) SURF1_NoMut lines that we generated by correcting the c.769G>A mutation in both alleles of patient S2 iPSCs (S2_Corr1, S2_Corr2). The SURF1 group included (i) SURF1_Mut lines that we derived from LS patients (S1, S2) and (ii) CTL_Mut lines that we obtained by introducing the c.769G>A mutation into both alleles of control C1 iPSCs (lines C1_Mut1, C1_Mut2) (Fig. 1d). We focused our further study on corrected lines S2_Corr1 and S2_Corr2, and on mutant line C1_Mut1.

We investigated the biochemical validity of our model by testing the assembly and functionality of COX in iPSC-derived NPCs. MT-CO2 protein was expressed in NPCs from healthy controls (CTL_NoMut: C1, C2), from human embryonic stem cells (hESCs: H1), and from genetically corrected iPSCs (SURF1_NoMut: S2_Corr1, S2_Corr2) (Fig. 1e). We could not detect MT-CO2 protein in patient-derived NPCs (SURF1_Mut: S1, S2) (Fig. 1e). The absence of a band in SURF1 NPCs suggested a rapid degradation of free MT-CO2 that was not assembled into COX. Blue-native gel electrophoresis (BNGE) confirmed the absence of MT-CO2 protein in the catalytic core of CIV in NPCs carrying the SURF1 mutation in a healthy background (CTL_Mut: C1_Mut1) (Fig. 1f). 2D BNGE confirmed that the introduction of the mutation into control NPCs (CTL_Mut: C1_Mut1) was sufficient to decrease the amount of fully-assembled CIV, leading to a partially assembled complex migrating at lower molecular weight (Fig. 1g, red arrows). SURF1 NPCs (CTL_Mut: C1_Mut1) also accumulated assembly intermediates containing MT-CO1 (Fig. 1g, red arrowhead). The impairment of CIV assembly prevented the formation of the supercomplex composed by complex III (CIII) and CIV (III2 + IV, detected by COX4I1 and UQCRC2 antibodies) (Fig. 1g, open red arrowheads), while the individual assembly of complex II (CII) and CIII was unaffected (Fig. 1g). The findings are in agreement with the reported functional defects of CIV assembly caused by SURF1 mutations[15,26].

We next assessed the enzyme activity of the COX complex using in situ enzyme analysis. We observed normal COX activity in NPCs from CTL (CTL_NoMut: C1, C2, C3; SURF1_NoMut: S2_Corr1, S2_Corr2). Conversely, COX activity was dramatically reduced and almost undetectable in SURF1 NPCs (SURF1_Mut: S1, S2; CTL_Mut: C1_Mut1) (Fig. 1h, i). The enzyme activity of CII (succinate dehydrogenase, SDH) was similar in all NPCs (Supplementary Fig. 2h). The findings confirmed that SURF1 mutations irrespective of the genomic background impair the activity of COX without affecting the activity of other complexes.

Taken together, the developed iPSC-based model of LS caused by SURF1 mutations recapitulated the disease defects at the biochemical level.

**SURF1 mutations impair neuronal maturation.** We set out to use the newly generated LS model to address the pathogenic effects of SURF1 mutations on the development of human neurons. We first employed a 2D differentiation protocol (Fig. 2a) generating differentiated neurons (DNs) enriched for dopaminergic tyrosine hydroxylase (TH)-positive neurons[39] (Supplementary Fig. 3a-b), whose degeneration contributes to the basal ganglia pathology of LS[40]. DN cultures contained also astrocytes (Supplementary Fig. 3c-d) that are an important CNS source of lactate, whose levels are increased in LS[4]. We compared SURF1 DNs (SURF1_Mut: S1, S2; CTL_Mut: C1_Mut1) to CTL DNs (CTL_NoMut: C1, C2, C3; SURF1_NoMut: S2_Corr1, S2_Corr2).

We used DNs derived from hESC line H1 to ensure that our control iPSCs were behaving similarly to the "gold standard" of pluripotent stem cells.

SURF1 DNs at 4 weeks (4w) and 8 weeks (8w) showed reduced numbers of TUJ1-positive neurons compared to control DNs (Fig. 2b). In agreement with the presence of COX impairment (Fig. 1h, i), SURF1 DNs had a reduction of oxygen consumption rate (OCR), maximal respiration, and ATP production rate compared to CTL DNs (Fig. 2c, d, Supplementary Fig. 3e–h). All DN cultures showed electrophysiological maturation (Supplementary Fig. 3j–l) and higher amount of secreted cytokines over time (Supplementary Fig. 3i). However, neuronal maturation appeared defective in SURF1 DNs, which exhibited reduced sodium and potassium currents at 4w and 8w compared to CTL DNs (Fig. 2e, f). SURF1 DNs also lacked repetitive spiking and postsynaptic currents (Fig. 2g), and contained higher numbers of non-spiking neurons (Supplementary Fig. 3m).

We next investigated whether neuronal maturation defects could be recapitulated in 3D. We derived cortical brain organoids[41,42], and analyzed them after 40 days (D40) and 90 days (D90), which may correspond to weeks 12 and 16 post-conception[43] (Fig. 2h and Supplementary Data 2). We compared SURF1 organoids (SURF1_Mut: S1, S2; CTL_Mut: C1_Mut1) to CTL organoids (CTL_NoMut: C1, C2; SURF1_NoMut: S2_Corr1, S2_Corr2) (Supplementary Data 2). The presence of TUJ1-positive and synaptophysin (SYP)-positive neurons was reduced in D40 SURF1 organoids compared to CTL organoids (Fig. 2i, Supplementary Fig. 3n), suggesting aberrant neuronal development in mutants. The defective neuronal maturation of mutant cells became more evident at D90, as seen by the further reduction of TUJ1-positive, MAP-positive, and SYP-positive neurons in SURF1 organoids compared to CTL organoids (Fig. 2i).

Collectively, 2D and 3D cultures indicated that SURF1 mutations disrupted human neuronal maturation.

**Single-cell transcriptomics identifies defective acquisition of mature neural fate in SURF1-mutant 2D and 3D cultures.** To elucidate the impact of SURF1 mutations on neuronal generation, we investigated the cellular composition of 2D and 3D cultures using droplet-based single-cell RNA-sequencing. We analyzed the single-cell transcriptome of 4w DNs from CTL (CTL_NoMut: C1; SURF1_NoMut: S2_Corr1) and SURF1 (SURF1_Mut: S2). We observed a clear separation between control and mutant populations in the uniform manifold approximation and projection (UMAP)[44] dimensionality reduction plot (Fig. 3a). K-nearest neighbors based clustering of DN-derived single cells identified seven clusters (Fig. 3b, Supplementary Data 3).

Control DNs populated clusters 1, 2, 3, 4, and 5 (Fig. 3b, c). Clusters 1 and 4 were enriched in cells with progenitor and glial identity expressing genes associated with progenitors (PTPRZ1, PTN) and astrocytes (SLC1A3, also known as GLAST1, SOX9, S100B, DLK1) (Fig. 3c, Supplementary Data 3). Clusters 2, 3, and 5 were composed of cells with gene signature related to neurite outgrowth (ROBO3, NEFM), control of proliferation (MEG3, MEIS2), and neuronal maturation (DCX, ANK2, NSG2, STMN2) (Fig. 3c, Supplementary Data 3).

SURF1 DNs populated clusters 0, 6, and 7 (Fig. 3b, c). Cluster 0 and 6 were enriched in cells expressing genes related to cell cycle and cancer-associated proliferation (SFRP2, CENPF, TOP2A, UBE2C, CRABP2, MYC) (Fig. 3c, Supplementary Data 3). Cluster 7 included cells with aberrant neuronal identity that expressed neuronal development genes (NEFM, DCX, STMN2, HES6) but also cell cycle and cancer-related genes (CRABP2, CCND1, MYC) (Fig. 3c, Supplementary Data 3). In accordance with a failed

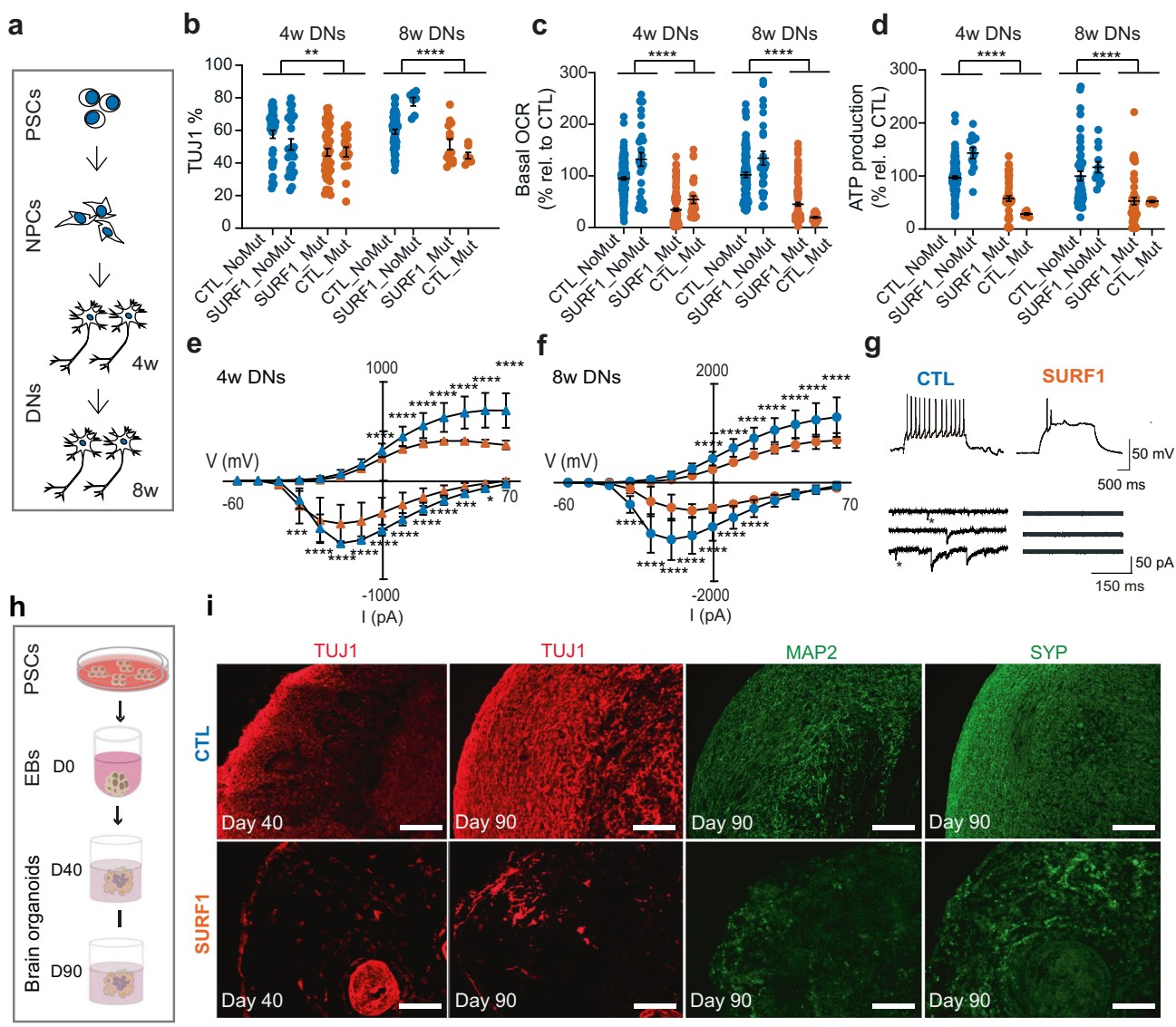

**Fig. 2 *SURF1* mutations impair neuronal maturation in 2D and 3D cultures. a** 2D differentiation into mature differentiated neurons (DNs). We analyzed DNs at 4 weeks (4w) and 8 weeks (8w) starting from NPCs. **b** HCA quantification of TUJ1 + neurons in 4w and 8w DNs from CTL (CTL_NoMut: C1, C2, C3; SURF1_NoMut: S2_Corr1, S2_Corr2) and SURF1 (SURF1_Mut: S1, S2; CTL_Mut: C1_Mut1) (mean ± s.e.m.; each dot represents a biological replicate; $n =$ 20 biological replicates per line over three independent experiments; **$p = 0.0013$, ****$p < 0.0001$ CTL vs. SURF1; two-sided Mann–Whitney U test). **c**, **d** Bioenergetics of 4w DNs and 8w DNs from CTL (CTL_NoMut: C1, C2, C3; SURF1_NoMut: S2_Corr1, S2_Corr2) and SURF1 (SURF1_Mut: S1, S2; CTL_Mut: C1_Mut1) (mean ± s.e.m.; each dot represents a biological replicate; $n =$ 20 biological replicates per line over three independent experiments; ****$p < 0.0001$ CTL vs. SURF1; two-sided Mann–Whitney U test). **e**, **f** Sodium (below $x$ axis) and potassium (above $x$ axis) currents in 4w and 8w DNs from CTL (CTL_NoMut: C1, C2, C3; SURF1_NoMut: S2_Corr1, S2_Corr2) and SURF1 (SURF1_Mut: S1, S2; CTL_Mut: C1_Mut1) (mean ± s.d.; $n =$ 40 individual cells per line over three independent experiments; ***$p < 0.001$, ****$p < 0.0001$ CTL vs. SURF1; one-way ANOVA followed by Bonferroni multiple comparison test). **g** Electrophysiology traces in current-clamp recordings for 8w DNs from CTL (SURF1_NoMut: S2_Corr1) and SURF1 (SURF1_Mut: S2). Above: spiking activity; below: spontaneous postsynaptic activity; stars: glutamatergic postsynaptic currents. **h** 3D differentiation into cerebral organoids. We counted days of differentiation starting from embryoid bodies (EBs). **i** Cerebral organoids from CTL (SURF1_NoMut: S2_Corr1) and SURF1 (SURF1_Mut: S2) at D40 and D90. Reproduced in CTL (CTL_NoMut: C1, C2) and SURF1 (SURF1_Mut: S1; CTL_Mut: C1_Mut1) (3–8 organoids per line per experiment, $n =$ 3 independent experiments). Scale bars: 100 μm.

acquisition of neuronal and glial fate, the distribution of cell cycle stages of SURF1 DNs was suggestive of enhanced proliferation, with higher number of cells in G2M and S2 phase compared to CTL DNs (Fig. 3d, e, Supplementary Fig. 4a). Accordingly, cells expressing proliferative markers (*MYC, TOP2A*) clustered mainly within *SURF1*-mutant population (Fig. 3f, g).

We next profiled the single-cell transcriptome of 3D brain organoids at D90 from CTL (CTL_NoMut: C1; SURF1_NoMut: S2_Corr1) and SURF1 (SURF1_Mut: S2). CTL and SURF1

organoids showed a clearly defined separation (Fig. 3h, Supplementary Fig. 4b). k-nearest neighbors based clustering of organoids-derived single cells identified 13 clusters (Fig. 3i, Supplementary Data 3).

CTL organoids populated clusters 2, 4, 6, 7, 8, 10, and 12 (Fig. 3i, j). Cluster 2, 8, and 12 contained cells with gene signature indicative of choroid plexus (*TTR, CXCL14*), hypothalamic and diencephalon neurons (*PMCH, PCP4, RSPO3*), apical radial glia (*ANXA1, CRYAB*), and glial cell fate (*GFAP, SOX9, NFIA, S100B*) (Fig. 3j,

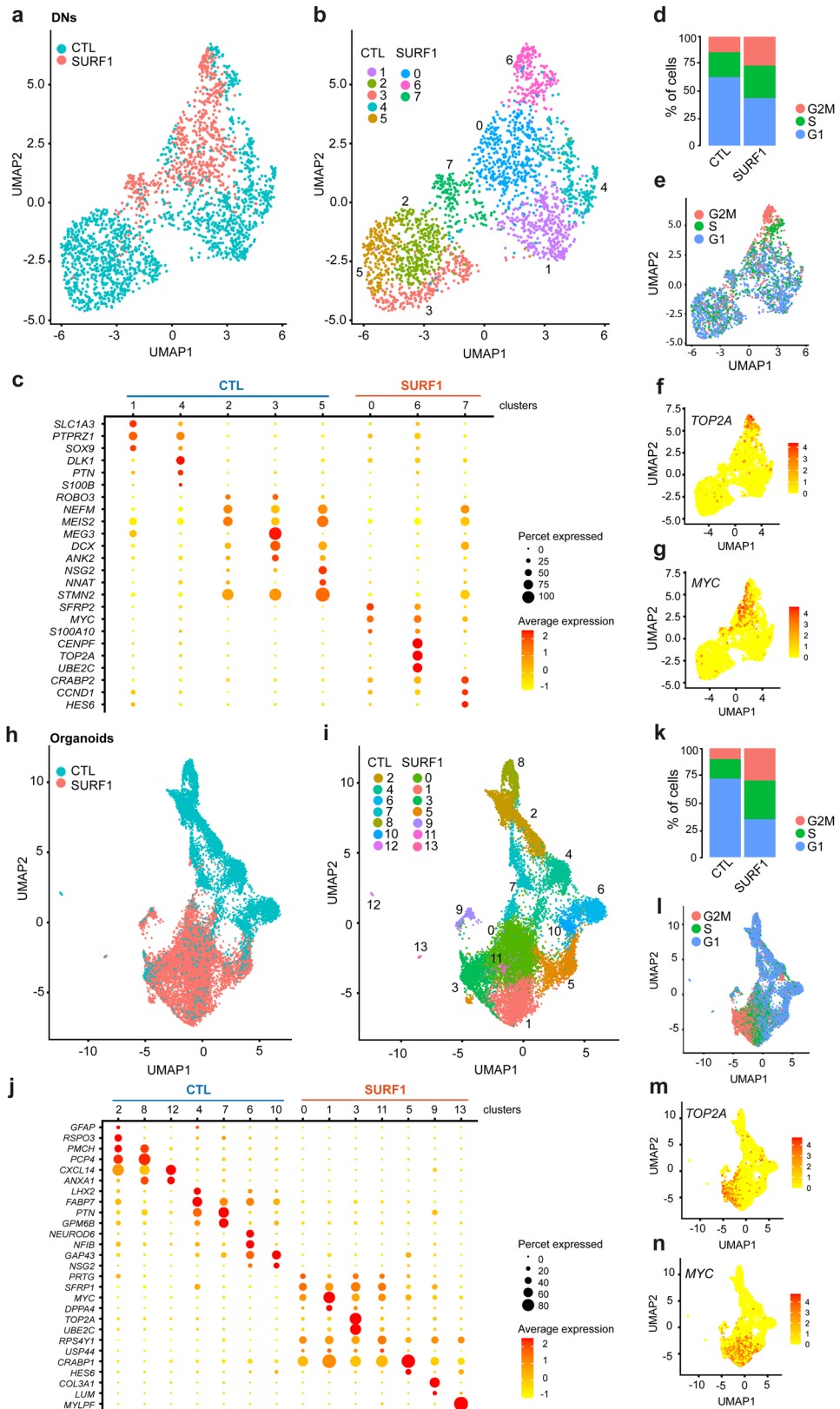

Supplementary Data 3). Cluster 4 and 7 comprised cells expressing genes associated with neural progenitors (*PAX6, LHX2, FABP7, PTN, GPM6B*) and outer radial glia (*FAM107A, PTPRZ1*) (Fig. 3j, Supplementary Data 3). Cluster 6 and 10 included cells expressing neuronal markers (*FOXG1, DCX, SMNT4, NFIB, NSG2, SYT1*) (Fig. 3j, Supplementary Data 3).

SURF1 organoids populated clusters 0, 1, 3, 5, 9, 11, and 13 (Fig. 3i, j). Cluster 0, 1, 3, and 11 contained cells expressing proliferative and cell cycle genes (*CRABP1, SFRP1, MYC, TOP2A, UBE2C, PRTG*), and pluripotency-associated genes (*LIN28A, DPPA4, POU5F1, USP44*) (Fig. 3j, Supplementary Data 3). Cells in cluster 5 resembled those in cluster 7 of DNs, and included cells

**Fig. 3 Single-cell transcriptomics highlights an imbalance between proliferation and maturation in *SURF1*-mutant neural cultures and brain organoids.**
**a** Uniform manifold approximation and projection (UMAP) plot showing the distribution of 4w DNs from CTL (CTL_NoMut: C1; SURF1_NoMut: S2_Corr1) and SURF1 (SURF1_Mut: S2) (*n* = 2 independent experiments). **b** UMAP plot of 4w DNs from CTL (CTL_NoMut: C1; SURF1_NoMut: S2_Corr1) and SURF1 (SURF1_Mut: S2) showing seven cellular clusters (resolution = 1) (*n* = 2 independent experiments). **c** Dot plot highlighting the expression of representative genes across clusters of DNs. Sizes of each dot reflect percentage of cells in the cluster where the gene is detected; colors reflect average expression level within each cluster (yellow: low expression; red: high expression). **d** Cell cycle distribution in 4w DNs from CTL (CTL_NoMut: C1; SURF1_NoMut: S2_Corr1) and SURF1 (SURF1_Mut: S2) (*n* = 2 independent experiments). **e** UMAP plot of 4w DNs from CTL (CTL_NoMut: C1; SURF1_NoMut: S2_Corr1) and SURF1 (SURF1_Mut: S2) showing cell cycle distribution (*n* = 2 independent experiments). **f, g** UMAP plots depicting signature distribution of cells expressing proliferative genes TOΠ2A and MΨX in 4w DNs from CTL (CTL_NoMut: C1; SURF1_NoMut: S2_Corr1) and SURF1 (SURF1_Mut: S2) (*n* = 2 independent experiments). **h** UMAP plot showing distribution of D90 organoids from CTL (SURF1_NoMut: S2_Corr1) and SURF1 (SURF1_Mut: S2) (3–8 organoids per line per experiment, *n* = 2 independent experiments). **i** UMAP plot of D90 organoids from CTL (SURF1_NoMut: S2_Corr1) and SURF1 (SURF1_Mut: S2) showing 13 cellular clusters (resolution = 0.6) (3–8 organoids per line per experiment, *n* = 2 independent experiments). **j** Dot plot expression of representative genes across clusters of 90D organoids. **k** Cell cycle distribution in 90D organoids from CTL (SURF1_NoMut: S2_Corr1) and SURF1 (SURF1_Mut: S2) (3–8 organoids per line per experiment, *n* = 2 independent experiments). **l** UMAP plot of 90D organoids from CTL (SURF1_NoMut: S2_Corr1) and SURF1 (SURF1_Mut: S2) showing cell cycle distribution (38 organoids per line per experiment, *n* = 2 independent experiments). **m, n** UMAP plots depicting signature distribution of cells expressing proliferative genes TOΠ2A and MΨX in 90D organoids from CTL (SURF1_NoMut: S2_Corr1) and SURF1 (SURF1_Mut: S2) (3–8 organoids per line per experiment, *n* = 2 independent experiments).

with aberrant neuronal identity that expressed neuronal genes (*GAP43, NEFM, DCX, HES6*) and cancer-related proliferative genes (*CRABP1, MYC, SFRP1*) (Fig. 3c, Supplementary Data 3). Cluster 9 and cluster 13 comprise non-neuronal cells expressing markers of neuroepithelial (*COL3A1, LUM*) and mesodermal identity (*MYLPF, TTN*) (Fig. 3c, Supplementary Data 3). Congruent with a failure of proper glial and neuronal maturation, cell cycle stage distribution of SURF1 organoids showed increased cells in G2M and S2 stages and fewer cells in G1 stage compared to CTL organoids (C1, S2_Corr1) (Fig. 3k, l, Supplementary Fig. 4a). SURF1 organoids were in fact enriched for cells expressing proliferative markers (*MYC, TOP2A*) (Fig. 3m, n).

Altogether, single-cell transcriptomics suggested that *SURF1* mutations disrupted neuronal maturation by causing an imbalance between proliferation and differentiation.

**Multi-omics analysis reveals aberrant control of metabolism, proliferation, and morphogenesis in SURF1 neural cultures.** We next aimed to gain insights into the mechanisms underlying the *SURF1* mutation-induced failure in controlling cell proliferation during neuronal generation. We carried out multi-omics analysis of 8w DNs from CTL (SURF1_NoMut: S2_Corr1) and SURF1 (SURF1_Mut: S2) (Fig. 4a). We integrated total RNA-sequencing (Supplementary Fig. 4c–d, Supplementary Data 5), proteomics (Supplementary Data 6-7), and metabolomics (Supplementary Data 8) using xMWAS, a software for integration of multiple omics platforms[45]. xMWAS identified three communities composed of tightly connected genes, protein, and metabolites (Fig. 4b, Supplementary Data 4).

Enriched pathways in community 1 were related to signaling and morphogens (Fig. 4b, Supplementary Data 4). Disrupted signaling pathways included TGFβ pathway (*TGFBR1, NODAL, LEFTY2, IGF2BP1*), WNT pathway (*DKK3, SFRP2, WNT7B*), and sonic hedgehog (SHH) (Fig. 4c, d).

Community 2 was enriched in metabolic pathways (community 2 contained 80 metabolic pathways, while community 1 contained 28 and community 3 contained 45) (Fig. 4b, Supplementary Data 4). Downregulated genes and proteins in SURF1 DNs involved mitochondrial bioenergetics (*MT-CO3, PDK4, PPARGC1A*) (Fig. 4c, d, Supplementary Fig. 4f). Downregulated proteomics-related pathways included electron transport chain (Supplementary Fig. 4h). Downregulated metabolites included carnitine, choline, xanthine, and energy-related molecules AMP, ADP, and ATP (Fig. 4e). We also observed signs of altered nicotinamide adenine dinucleotide (NAD) metabolism in SURF1 DNs, as shown by low amount of NAD and NADP (Fig. 4e) and low expression of enzymes involved

in NAD and NADP metabolism (*NAMPT, GPD1, G6PD1*) (Fig. 4c, d, Supplementary Data 5). Interestingly, low NADP was identified as part of the metabolic signature of LS[46] and exogenous NAD+ rescued the cell death induced by glucose-by-galactose replacement in fibroblasts derived from LS patients carrying CI mutations[47]. Within community 2, we also identified increased expression of regulators of cell proliferation (*CENPE, MKI67, TOP2A, UBE2C,* guanine), cancer, and pluripotency (*DNMT3B, DPPA4, MYC, POU5F1, CRABP2*) (Fig. 4c–e). Upregulated proteomics-related pathways in SURF1 DNs included DNA replication (Supplementary Fig. 4h), and DNA and chromosome-related cell compartments (Supplementary Fig. 4g). Upregulated metabolites included glycolysis-related metabolites, like D-fructose 2,6-bisphosphate (2FDP) and 2-phosphodiglyceric acid (2PDG) (Fig. 4e).

Community 3 comprised regulators of neuronal function that were downregulated in SURF1 DNs (Fig. 4b, Supplementary Data 4), including markers of neuronal morphogenesis and axon guidance (*ALCAM, CHL1, EFNB3, GAS7, RELN, SEMA3C, SLIT2, SPON1*), and markers of glial and neuronal development (*CDH6, CD44, GFAP, NCAM2, SYNPO*) (Fig. 4c, d, Supplementary Fig. 4e). Downregulated proteomics-related pathways in community 3 included cell compartments related to "cell projection" and "synapse" (Supplementary Fig. 4g) and biological processes regulating "neuron project guidance" (Supplementary Fig. 4h). Decreased metabolites in SURF1 DNs included CNS-specific metabolites such as N-acetyl aspartyl glutamate (NAG) and N-acetyl aspartate (NAA), which is synthesized in mitochondria and is known to be reduced in LS and neurological disorders[9,48] (Fig. 4e). Community 3 metabolites upregulated in SURF1 DNs included folate and cAMP, which are important in neuronal development and promotion of axonal growth[49]. The increase of folate and cAMP, which were provided to DNs as part of their medium, might suggest that they were not effectively metabolized by mutant cells. Hence, impaired neuronal morphogenesis caused by *SURF1* mutations may prevent the cells to utilize the available neurodevelopmental-promoting cues.

Overall, multi-omics analysis of SURF1 neural cultures suggested that metabolic defects caused by *SURF1* mutations constrained differentiating cells to retain a proliferative and glycolytic state, which failed to support neuronal morphogenesis and maturation.

**LS-associated defects in metabolism, proliferation, and morphogenesis emerge at the level of NPCs.** We sought to validate the occurrence of impaired neuronal morphogenesis in SURF1 cultures. We developed a high-content analysis (HCA) assay for

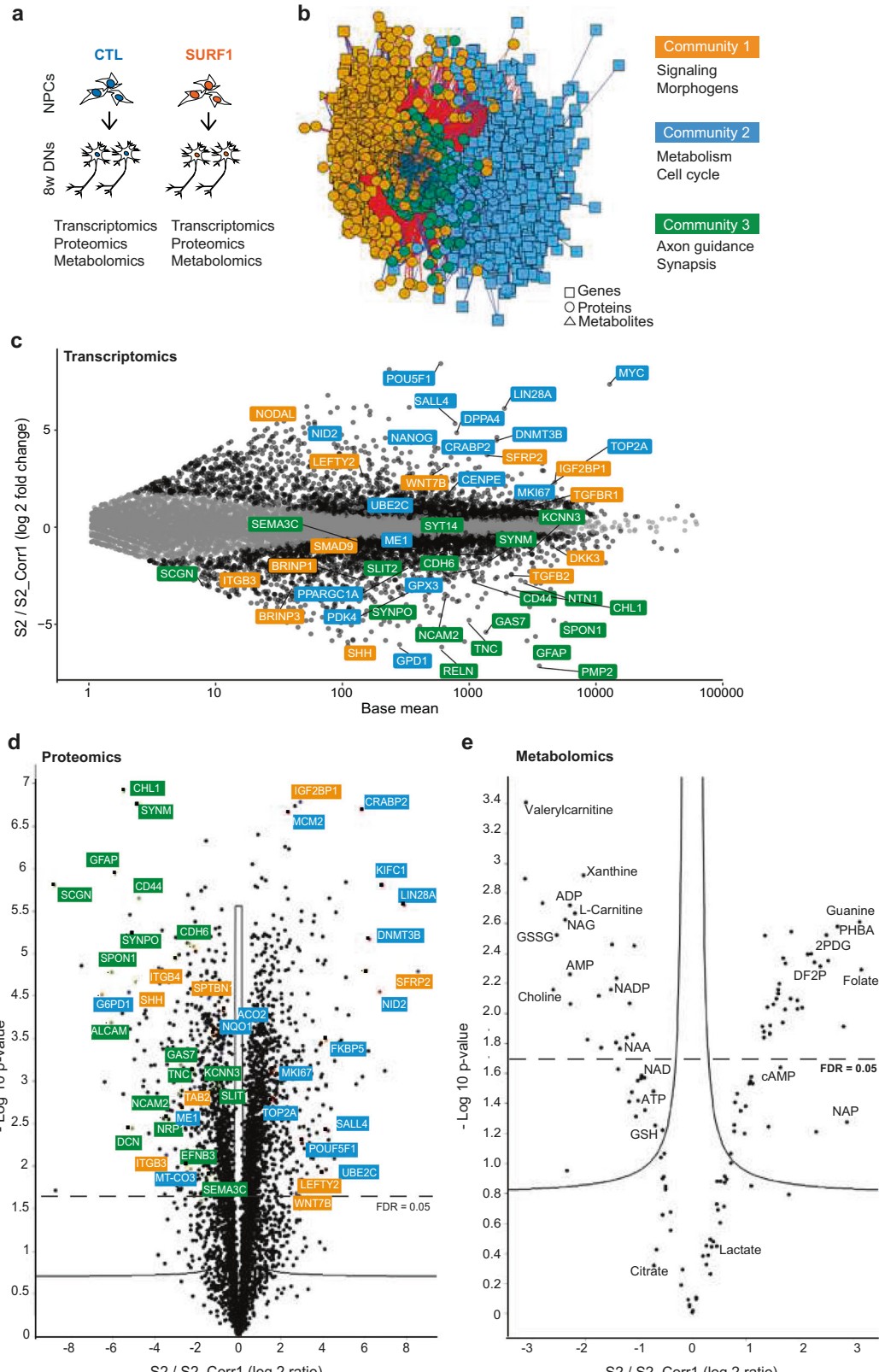

quantitative assessment of neurite outgrowth and branching complexity based on TUJ1-positive neuronal cells. We assessed 4w and 8w DNs from SURF1 (SURF1_Mut: S1, S2; CTL_Mut: C1_Mut1) and CTL (CTL_NoMut: C1, C2, C3; SURF1_NoMut: S2_Corr1, S2_Corr2). We used DNs derived from hESC line H1 as a baseline for CTL DNs.

SURF1 DNs at 4w and 8w exhibited significantly reduced neurite length and number of branch points in comparison to CTL DNs (Fig. 5a–c). Thus, *SURF1* mutations impaired neuronal morphogenesis not only in mature neurons (8w DNs) but also in less mature neurons (4w DNs) (Fig. 5b, c). In fact, even if the transcriptome of SURF1 DNs and CTL DNs diverged mostly

**Fig. 4 Multi-omics analysis of SURF1 neural cultures identifies dysregulation of morphogens, metabolism/proliferation, and neuronal wiring.** **a** Schematics of multi-omics of 8w DNs from CTL (SURF1_NoMut: S2_Corr1) and SURF1 (SURF1_Mut: S2). **b** Multi-omics integration performed with xMWAS identified three communities. Genes are indicated by squares, proteins by circles, and metabolites by triangles. **c** MA plot showing log fold changes (LFC) of differentially expressed genes in 8w DNs from SURF1 (SURF1_Mut: S2) compared to CTL (SURF1_NoMut: S2_Corr1). Highlighted genes are shown using colors of their respective community ($n = 3$ independent experiments). **d** Volcano plot depicting differentially expressed proteins in 8w DNs from SURF1 (SURF1_Mut: S2) compared to CTL (SURF1_NoMut: S2_Corr1), after Benjamini–Hochberg correction for multiple testing. Highlighted proteins are shown using colors of their respective community ($n = 3$ independent experiments). **e** Volcano plot depicting differentially detected metabolites in 8w DNs from SURF1 (SURF1_Mut: S2) compared to CTL (SURF1_NoMut: S2_Corr1) after Benjamini–Hochberg correction for multiple testing ($n = 3$ independent experiments). 2FDP: D-fructose 2,6-bisphosphate; 2PG: 2-phosphoglyceric acid; GSSG: oxidized-glutathione; GSH: reduced-glutathione; NADP: nicotinamide adenine dinucleotide phosphate; NAA: N-acetyl aspartate; NAG: N-acetyl aspartyl glutamate.

at 8w of neuronal differentiation, transcriptomic differences between mutant and control DNs were already evident at 4w (Supplementary Fig. 4c-d, Supplementary Data 9-10). These results collectively indicated that disease defects might emerge early during neurogenesis.

Next, we investigated whether iPSC-derived NPC cultures (Supplementary Fig. 5a) would already exhibit disease phenotypes. We had hints that SURF1 NPCs may be defective: 4w and 8w SURF1 DNs contained fewer NCAM-positive neuron-restricted neural progenitors (Supplementary Fig. 5b), and the SURF1-specific transcriptomic signature of SURF1 DNs and SURF1 organoids was already, at least in part, recapitulated by SURF1 NPCs (Supplementary Fig. 5c). We therefore compared SURF1 NPCs (SURF1_Mut: S1, S2; CTL_Mut: C1_Mut1) to CTL NPCs (CTL_NoMut: C1, C2, C3; SURF1_NoMut: S2_Corr1, S2_Corr2). SURF1 NPCs exhibited mitochondrial bioenergetic defects with reduced basal OCR, maximal respiration, and ATP production compared to CTL NPCs (Fig. 5g, h, Supplementary Fig. 5d). At the same time, SURF1 NPCs retained a stronger glycolytic profile than CTL NPCs, with higher extracellular acidification rate (ECAR) and increased lactate release (Fig. 5i, j), which is a hallmark of LS[4]. The findings are in accord with the upregulation of glycolysis in SURF1 DNs suggested by multi-omics (Fig. 4c–e).

We addressed whether the metabolic defects of NPCs could be recapitulated in non-neuronal cells. During the diagnostic work-up, fibroblasts from SURF1 patients (SURF1_Mut: S1, S2) were found to exhibit COX deficiency (see subject details in the "Methods" section). However, when we measured the bioenergetic profile of SURF1 fibroblasts (SURF1_Mut: S1, S2) compared to control fibroblasts (CTL_NoMut: C2, C3), we did not detect significant changes in their basal OCR or ATP production rate, despite a slightly reduced maximal respiration rate (Fig. 5i, Supplementary Fig. 5e). These results confirmed the known tissue-specificity of SURF1[26,27], and suggest that the consequences of COX deficiencies caused by SURF1 mutations may be different in distinct cell types. Accordingly, SURF1 fibroblasts did not show glycolytic upregulation; their ECAR levels were even lower than those of control fibroblasts, and their lactate release remained unchanged (Fig. 5j, k). We also failed to observe significant changes in the mitochondrial or glycolytic profile of undifferentiated SURF1 iPSCs (SURF1_Mut: S2) compared to control iPSCs (CTL_NoMut: C1; SURF1_NoMut: S2_Corr1, S2_Corr2) (Supplementary Fig. 5f-h). Altogether, SURF1 mutations appeared to cause an impairment of energy metabolism that affected neuronal cells more dramatically (Fig. 2c, d), and that became evident already at the level of NPCs (Fig. 5d–h).

We next addressed the consequences of these metabolic defects on NPC function. We compared SURF1 NPCs (SURF1_Mut: S1, S2; CTL_Mut: C1_Mut1) to CTL NPCs (CTL_NoMut: C1, C2, C3; SURF1_NoMut: S2_Corr1, S2_Corr2). SURF1 NPCs expressed higher levels of proliferative and pluripotency-associated markers (c-MYC, OCT4) (Fig. 5l, m). SURF1 NPCs also proliferated at

higher rates compared to CTL NPCs (Fig. 5n). NPCs thus recapitulated the increase of proliferative and pluripotency-associated markers and the cell cycle changes observed in SURF1 DNs (Fig. 3c–g, and 4c, d) and SURF1 organoids (Fig. 3j–n). SURF1 NPCs exhibited significantly elevated mtDNA copy numbers (Fig. 5o), possibly as an attempt to compensate for the energy deficiency by increasing the mitochondrial mass. Nonetheless, the metabolic alterations of SURF1 NPCs did not affect the production of mitochondrial reactive oxygen species (ROS) (Supplementary Fig. 5i-j). Mitochondrial ROS in SURF1 NPCs remained similar to those in CTL NPCs, both at the basal level and after mitochondrial stress (Supplementary Fig. 5i, j). SURF1 mutations also did not alter the ultrastructural morphology of mitochondria in NPCs (Supplementary Fig. 5k). Hence, the defective metabolism of NPCs did not cause widespread mitochondrial alterations.

We next assessed the state of neuronal morphogenesis of immature early neurons that are present within NPC cultures. Although the branching length of these TUJ1-positive immature neurons was small (Fig. 5p), we could detect significantly decreased neurite outgrowth (neurite length and branching points) in SURF1 NPCs compared to CTL NPCs (Fig. 5p–r). We asked whether such neural morphogenesis defects could be observed in other LS-causing mutations. We generated iPSCs from two LS patients carrying mutations in the nuclear CI gene NDUFS4 (Supplementary Fig. 5l). The first patient (NDU_1) carried the mutation c.462delA p.(K154fs) and the second patient (NDU_2) carried the mutation c.316C>T p.(R106*). We compared NPCs derived from iPSCs carrying NDUFS4 mutations (NDUFS4_Mut: NDU_1, NDU_2) to CTL NPCs (CTL_NoMut: C1, C2, C3). NDUFS4 NPCs exhibited the expected reduction in mitochondrial membrane potential (MMP) typical of CI defects[6,34] (Fig. 5s). TUJ1-positive neurons within NDUFS4 NPCs showed reduced neurite outgrowth compared to CTL NPCs (Fig. 5t, u). These findings indicate that early neuronal morphogenesis defects might potentially represent a general pathogenetic mechanism of LS that is not restricted to SURF1 mutations.

Overall, SURF1 mutations prevented NPCs from shifting toward OXPHOS, thereby promoting a proliferative and glycolytic state associated with an insufficient lack of guidance clues to drive neuronal morphogenesis.

**Disrupted neuronal progenitor cytoarchitecture in SURF1-mutant cerebral organoids.** We aimed to address whether 3D cerebral organoids (Supplementary Fig. 6a) could recapitulate the disruption of NPC function and early morphogenesis observed in 2D. We first carried out total RNA-sequencing of D90 brain organoids from CTL (SURF1_NoMut: S2_Corr1) and SURF1 (SURF1_Mut: S2) (Fig. 6a, Supplementary Data 11). The expression pattern of genes belonging to the three communities identified by xMWAS agreed with the results from the 2D analysis (Fig. 6a, Supplementary Data 11). For genes in community 1, SURF1 organoids showed deregulated expression of

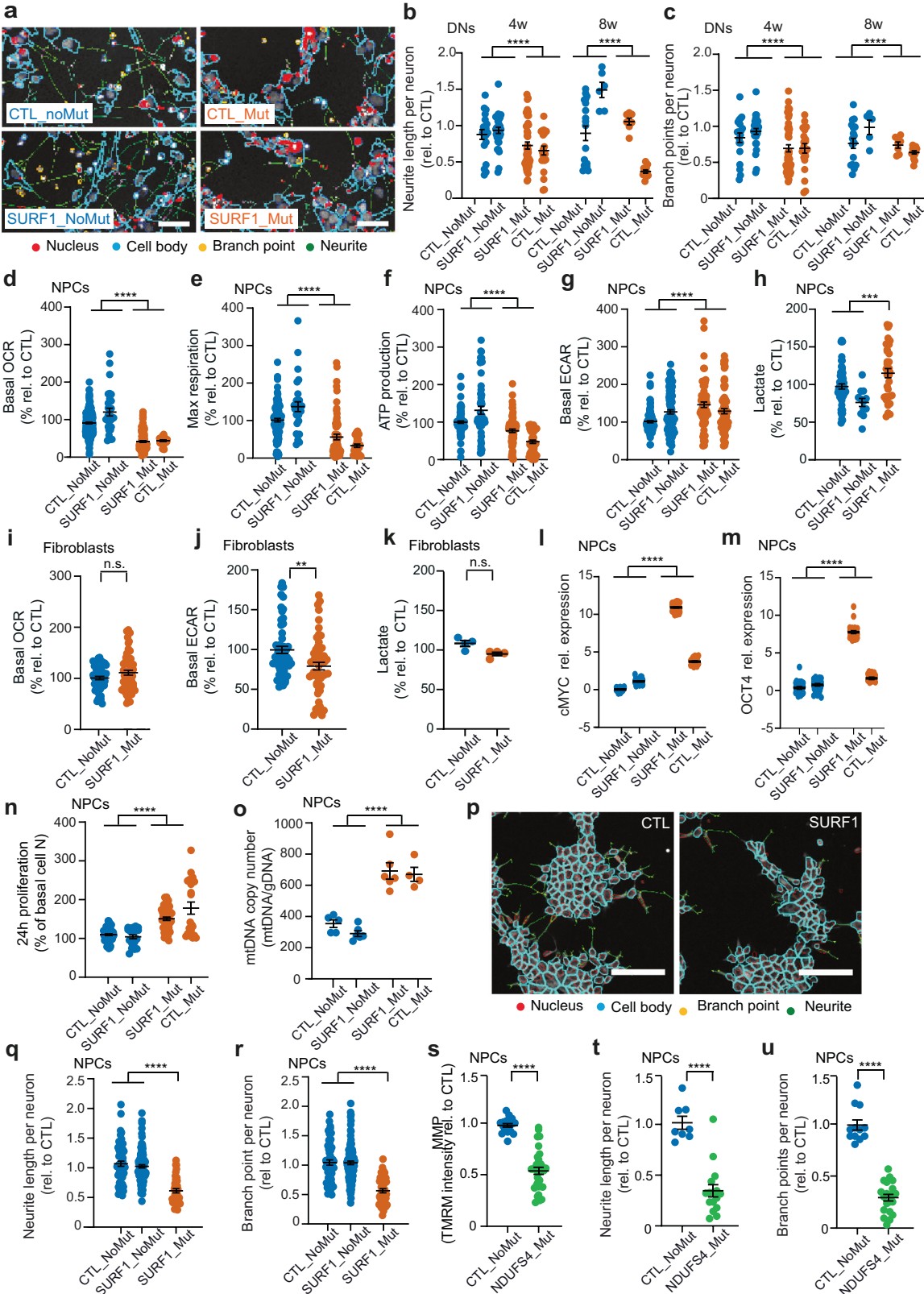

morphogens, including TGFβ (*BMP4, TGFB2, IGF2BP1*) and WNT pathway (*SFRP2, WNT2B*) (Fig. 6a, Supplementary Data 11). For genes in community 2, SURF1 organoids downregulated bioenergetic genes (*PDK4, PPARGC1A*) (Fig. 6a, Supplementary Fig. 6b, Supplementary Data 11), and upregulated glycolytic, proliferative, and pluripotency-associated genes (*MKI67, CENPE, CRABP2, MYC, LDHA, POU5F1*) (Fig. 6a, Supplementary Fig. 6b, Supplementary Data 11). Genes of community 3 related to neuronal function were mainly downregulated in SURF1 organoids, including axon guidance genes (*CHL1, GAS7, RELN, SEMA3C, SLIT2*), and glial and neuronal development genes and synaptic genes (*SYT14, CDH6, FOXG1,*

**Fig. 5 LS-associated defects in neuronal morphogenesis emerge at the level of NPCs. a–c** HCA masks and neuronal morphogenesis of 4w and 8w DNs from CTL (CTL_NoMut: C1, C2, C3; SURF1_NoMut: S2_Corr1) and SURF1 (SURF1_Mut: S1, S2: CTL_Mut: C1_Mut1) (mean ± s.e.m.; $n = 20$ biological replicates (dots) per line over three independent experiments; ****$p < 0.0001$ CTL vs. SURF1; two-sided Mann–Whitney U test). Scale bar: 50 μm. **d–h** Bioenergetic profile and lactate release in NPCs from CTL (CTL_NoMut: C1, C2, C3; SURF1_NoMut: S2_Corr1) and SURF1 (SURF1_Mut: S1, S2; CTL_Mut: C1_Mut1) (mean ± s.e.m.; $n = 20$ biological replicates (dots) per line over three independent experiments; ***$p < 0.001$, ****$p < 0.0001$ CTL vs. SURF1; two-sided Mann–Whitney U test). **i–k** Bioenergetic profile and lactate release in fibroblasts from CTL (CTL_NoMut: C2, C3) and SURF1 (SURF1_Mut: S1, S2) (mean ± s.e.m.; $n = 15$ biological replicates (dots) per line (**i**, **j**) and $n = 4$ biological replicates (dots) per line (**k**) over two independent experiments; n.s. = not significant, **$p < 0.01$ CTL vs. SURF1; two-sided Mann–Whitney U test). **l**, **m** QRT-PCR in NPCs from CTL (CTL_NoMut: C1, C2, C3; SURF1_NoMut: S2_Corr1) and SURF1 (SURF1_Mut: S1, S2; CTL_Mut: C1_Mut1) (normalized to AXTB; mean ± s.e.m.; $n = 10$ replicates per line over two independent experiments; ****$p < 0.0001$ CTL vs. SURF1; two-sided Mann–Whitney U test). **n**, **o** 24 h proliferation and mtDNA quantification (normalized over genomic DNA, gDNA) in NPCs from CTL (CTL_NoMut: C1, C2, C3; SURF1_NoMut: S2_Corr1) and SURF1 (SURF1_Mut: S1, S2; CTL_Mut: C1_Mut1) (mean ± s.e.m.; $n = 20$ biological replicates (dots) per line (**n**) and $n = 4$ biological replicates (dots) per line (**o**) over three independent experiments; ****$p < 0.0001$ CTL vs. SURF1; two-sided Mann–Whitney U test). **p–r** HCA masks and neuronal morphogenesis in NPCs from CTL (CTL_NoMut: C1, C2, C3; SURF1_NoMut: S2_Corr1) and SURF1 (SURF1_Mut: S1, S2) (mean ± s.e.m.; $n = 20$ biological replicates (dots) per line over three independent experiments; ****$p < 0.0001$ CTL vs. SURF1; two-sided Mann–Whitney U test). Scale bar: 50 μm. **s–u** Mitochondrial membrane potential (MMP) quantification and morphogenesis in NPCs from CTL (CTL_NoMut: C2, C3) and NDUFS4 (NDUFS4_Mut: NDU_1, NDU_2) (mean ± s.e.m.; $n = 4$ biological replicates (dots) per line over three independent experiments; ****$p < 0.0001$ CTL vs. NDUFS4; two-sided Mann–Whitney U test).

*GFAP, LHX2*) (Fig. 6a, Supplementary Fig. 6b, Supplementary Data 11). These findings are in agreement with the observed reduction of SYP-positive and synapsis 1 (SYN)-positive neurons in SURF1 organoids compared to CTL organoids (Fig. 2i, Supplementary Fig. 3n, Supplementary Fig. 6c).

We next investigated the state of neural progenitors within brain organoids from SURF1 (SURF1_Mut: S1, S2; CTL_Mut: C1_Mut1) and CTL (CTL_NoMut: C1, C2; SURF1_NoMut: S2_Corr1). D40 CTL organoids contained LHX2-positive, SOX2-positive, and PAX6-positive neural progenitor populations that formed a well-organized architecture (Fig. 6b, Supplementary Fig. 6d), and a tightly packed neuroepithelial layer reminiscent of embryonic ventricles (Fig. 6c, white arrows). In contrast, D40 SURF1 organoids exhibited a disorganized neural progenitor pattern of expression (Fig. 6b–c, Supplementary Fig. 6d), and a disruption of neuroepithelial layers (Fig. 6c, white arrowhead). The spatial organization of p-VIM-positive cells, indicating dividing progenitors at subapical positions[50], was lost in D40 SURF1 organoids (Fig. 6b). At D90, CTL organoids still exhibited organized SOX2-positive progenitors, while this population was almost undetectable in SURF1 organoids (Fig. 6c, Supplementary Fig. 6e). The findings imply that increased progenitor proliferation might lead to the exhaustion of the NPC pool over time. In agreement with lack of organized progenitor growth and impaired neuronal differentiation, SURF1 organoids showed an overall reduced size (Fig. 6d, Supplementary Fig. 6f). SURF1 organoids also exhibited impaired OXPHOS function, with a decrease of OCR, maximal respiration, and ATP production rate compared to CTL organoids (Fig. 6e–g).

Altogether, the aberrant cytoarchitecture of 3D brain organoids confirmed the findings detected in 2D neural cultures, pointing toward a dysregulation of NPC function and morphogenesis as a mechanistic defect underlying SURF1-related pathogenesis.

**Bezafibrate treatment and *SURF1* gene augmentation enable the metabolic shift of NPCs and restore early morphogenesis.** We reasoned that if the neuronal defects of LS stemmed from a loss of NPC function, providing a healthy copy of the *SURF1* gene (without altering the presence of the mutation) could restore neuronal morphogenesis by enhancing OXPHOS. We first used lentiviruses to deliver wild-type (WT)-SURF1 to either SURF1 NPCs (SURF1_Mut: S1, S2) or 4w SURF1 DNs (SURF1_Mut: S1, S2) (Supplementary Fig. 7a). WT-SURF1 improved the bioenergetics of SURF1 NPCs (Fig. 7a and Supplementary Fig. 7b-c) and the neuronal morphogenesis of SURF1 DNs (Fig. 7b, Supplementary Fig. 7d). SURF1-WT also lowered the lactate production

of SURF1 DNs (Fig. 7c). We then used adeno-associated viruses (AAV), which is a system currently considered for in vivo gene transfer of CNS diseases[51], to deliver WT-SURF1 to either SURF1 NPCs (SURF1_Mut: S1, S2) or 4w SURF1 DNs (SURF1_Mut: S1, S2) (Supplementary Fig. 7e). Similar to lentiviruses, AAV delivery of WT-SURF1 ameliorated the bioenergetics of SURF1 NPCs (Fig. 7d, Supplementary Fig. 7f–h) and the morphogenesis of SURF1 DNs (Fig. 7e–f). These results collectively suggest that it may be possible to overcome *SURF1* mutation-specific neuronal defects with strategies that enhanced SURF1 function without eliminating the causative mutations.

We then assessed the extent of recovery following therapeutic strategies that have been proposed for LS. Hypoxia was beneficial in animal models of LS caused by knock-out of the CI gene *Ndufs4*[52,53]. We exposed SURF1 NPCs (SURF1_Mut: S1, S2) to hypoxia (5% oxygen) overnight (o.n.). Hypoxia improved mitochondrial bioenergetics of SURF1 NPCs (Supplementary Fig. 7i-k). However, hypoxia was not beneficial for NPC morphogenesis, as it reduced neuronal outgrowth in CTL NPCs and SURF1 NPCs (Supplementary Fig. 7l-m). The lack of improvement on neuronal morphogenesis might be due to a failure to support NPC to shift from glycolysis to OXPHOS. In fact, hypoxia increased ECAR and lactate concentration in SURF1 NPCs (Supplementary Fig. 7n-o). We observed similar results in SURF1 DNs (SURF1_Mut: S1, S2) exposed to o.n. hypoxia, which showed higher lactate production and failed to improve neuronal outgrowth (Supplementary Fig. 7p–r).

Next, we investigated the effects of antioxidants[4,54]. We treated SURF1 NPCs (SURF1_Mut: S1, S2) o.n. with n-acetyl-cysteine (NAC), alpha-tocotrienol (AT3), whose in vivo-derived metabolite EPI-743 has been suggested as a therapy for LS[55], ascorbic acid (AA), and dihydrolipoic acid (DHLA). Although some antioxidants lowered ECAR in SURF1 NPCs (Supplementary Fig. 7u), overall antioxidants failed to improve mitochondrial bioenergetics and morphogenesis of SURF1 NPCs (Supplementary Fig. 7t-v). Metabolic manipulations including increasing doses of glucose or pyruvate supplementation, previously proposed as treatment for LS caused by COX defects[56], also failed to promote OXPHOS and neuronal morphogenesis in SURF1 NPCs (SURF1_Mut: S1, S2) (Supplementary Fig. 7w-y).

Lastly, we investigated the potential benefits of pharmacologically activating mitochondrial biogenesis in SURF1 neural cells. SURF1 DNs (SURF1_Mut: S1, S2) and SURF1 organoids (SURF1_Mut: S1, S2) expressed reduced levels of *PPARGC1A*, which encodes for the master regulators of mitochondrial biogenesis PGC1A[57] (Supplementary Fig. 4f, Supplementary Fig. 6b). We confirmed

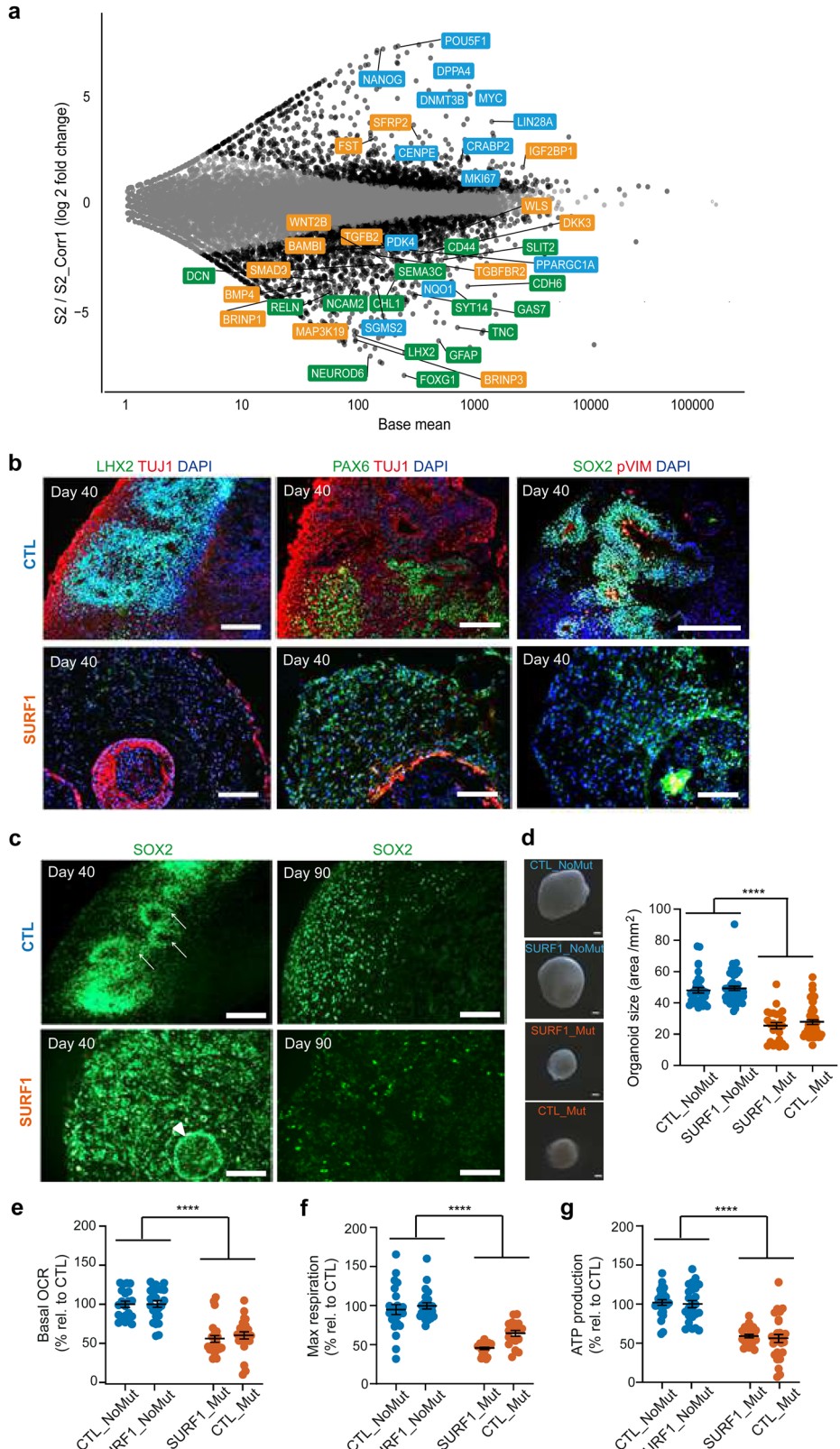

that PGC1A protein levels were low in SURF1 NPCs (SURF1_Mut: S1, S2; CTL_Mut: C1_Mut1) compared to CTL NPCs (CTL_NoMut: C1, C2; SURF1_NoMut: S2_Corr1) (Fig. 7g, h). To activate PGC1A in SURF1 NPCs (SURF1_Mut: S1, S2; CTL_Mut: C1_Mut1), we first used the peroxisome proliferator-activated receptor (PPAR) agonist bezafibrate (BZ), which has been suggested

for the treatment of neurological and mitochondrial disorders[58,59]. Treatment of SURF1 NPCs with 400 μM BZ o.n. increased PGC1A protein content (Fig. 7g, i) and elevated mtDNA copy number (Fig. 7j). BZ treatment in SURF1 NPCs also lowered the expression of proliferative and pluripotency-associated markers (c-MYC, OCT4) (Fig. 7k, l), and reduced cellular proliferation (Fig. 7m).

**Fig. 6 Aberrant cytoarchitecture and neural progenitor organization in SURF1 brain organoids. a** MA plot showing LFC of differentially expressed genes in D90 brain organoids from CTL (SURF1_NoMut: S2_Corr1) and SURF1 (SURF1_Mut: S2) (3–8 organoids per line per experiment, n = 3 independent experiments). Highlighted genes in community colors (orange: signaling and morphogens; blue: metabolism and cell cycle; green: axon guidance and synapsis). **b, c** Cerebral organoids from CTL (SURF1_NoMut: S2_Corr1) and SURF1 (SURF1_Mut: S2) at D40 and D90. White arrows indicate SOX2-positive neuroepithelium surrounding embryonic-like ventricles; white arrowhead indicates abnormal structure of neuroepithelium layering. Reproduced in CTL (CTL_NoMut: C1, C2) and SURF1 (SURF1_Mut: S1; CTL_Mut: C1_Mut1) (3–8 organoids per line per experiment, n = 3 independent experiments). Scale bar: 100 μm. **d** Representative images and quantification of the size of D90 organoids from CTL (CTL_NoMut: C1, C2; SURF1_NoMut: S2_Corr1) and SURF1 (SURF1_Mut: S2; CTL_Mut: C1_Mut1) (mean ± s.e.m.; 20–30 organoids measured per line per experiment; n = 5 independent experiments; ****p < 0.0001 CTL vs. SURF1; two-sided Mann–Whitney U test). **e–g** Bioenergetic profile of D40 organoids from (CTL_NoMut: C1, C2; SURF1_NoMut: S2_Corr1) and SURF1 (SURF1_Mut: S2, C1_Mut1; CTL_Mut: C1_Mut1) (mean ± s.e.m.; 5–15 organoids per line per experiment; n = 2 independent experiments; ****p < 0.0001 CTL vs. SURF1; two-sided Mann–Whitney U test).

Increasing doses of BZ treatment enhanced OXPHOS metabolism in SURF1 NPCs (Fig. 7n–p), and reduced glycolytic metabolism (Fig. 7q). Consequently, BZ improved the morphogenesis of SURF1 NPCs (Fig. 7r, Supplementary Fig. 7s). Lentivirus-mediated over-expression of PGC1A in SURF1 NPCs (SURF1_Mut: S1, S2; CTL_Mut: C1_Mut1) also improved OXPHOS bioenergetics and neuronal morphogenesis (Fig. 7s, t). These latter results confirmed that increased PGC1A expression might be the mechanism responsible for the beneficial effect of BZ treatment on SURF1 NPCs.

In summary, the loss of NPC function caused by *SURF1* mutations could be overcome (i) through viral-based gene augmentation of WT-SURF1 or (ii) through the increase of mitochondrial biogenesis via PGC1A induction that can be elicited by bezafibrate treatment (Supplementary Fig. 8). These two strategies enabled NPCs to shift toward oxidative metabolism and thereby supported neuronal morphogenesis.

## Discussion

*SURF1* mutations are among the most frequent monogenic defects causing LS, an incurable mitochondrial disease in children for which we lack a mechanistic understanding due to the paucity of effective model systems[6]. We discovered that *SURF1* mutations cause human neuronal impairment as a result of defective metabolic programming of NPCs that prevents the establishment of neuronal morphogenesis.

During neurogenesis, there is a shift from glycolysis to oxidative mitochondrial metabolism[60–62]. NPCs may represent the first cell type in this process to depend on OXPHOS function[31,63,64]. For this reason, OXPHOS-related defects might already emerge at the level of NPCs. Due to respiratory defects caused by *SURF1* mutations, SURF1 NPCs failed to undergo the metabolic shift, and retained glycolytic and proliferative features, which in turn hampered the establishment of neuronal morphogenesis. *SURF1* mutations did not lead to redox damage or widespread alterations of mitochondrial morphology, but caused metabolic defects that were rather specific to neuronal lineage cells. iPSCs and fibroblasts carrying the same *SURF1* mutations did not show the dramatic phenotypes observed in neural cells. Patient fibroblasts had isolated COX deficiency, but did not respond to this primary defect with a strong reduction of mitochondrial bioenergetics and upregulation of glycolysis. Perhaps this phenomenon is related to the known species-specific and tissue-specific effects of SURF1 on the assembly of COX[26,27]. In fact, COX has been suggested to contribute to the establishment of the transcriptional neuronal network[65] and to possibly represent an endogenous metabolic marker for neuronal activity[65,66]. Hence, improper COX assembly might lead to more dramatic consequences in neural cells.

In 3D cultures, dysfunctional NPCs disrupted the physiological neuronal layering causing aberrant cytoarchitecture. At later time points, brain organoids exhibited NPC exhaustion with impaired neuronal generation and reduced overall organoid size.

Mitochondrial bioenergetics is crucial for axonal arborization and synaptic function[67,68]. Neuronal wiring is crucial during brain development[69]. A failure to establish the orchestrated developmental events responsible for proper neuronal wiring and guidance can affect synaptogenesis and neuronal circuit formation[70]. These defects could underpin the cognitive and developmental impairment of LS patients and their susceptibility to metabolic disturbances that ultimately lead to neuronal cell dysfunction and death[71].

The genetic causes of LS are quite heterogeneous[5], but key pathological features are relatively conserved[40]. Therefore, the mechanisms identified for *SURF1* mutations might be potentially representative of general LS pathogenesis. Accordingly, we confirmed the presence of defective neurite outgrowth also in NPCs carrying mutations in the CI gene *NDUFS4*, another well-known cause of LS[34]. Previous studies demonstrated that NPCs carrying other LS-associated mutations showed bioenergetic defects[28] and mitochondrial calcium dyshomeostasis[29,31]. Hence, NPCs may play an unexpected role in the pathophysiology of LS. Since NPCs are not specific to the dopaminergic lineage, it is possible that several neuronal subtypes may be affected in LS. Accordingly, cerebral organoids from LS patients carrying *MT-ATP6* mutations displayed impaired maturation of cortical neurons, leading to an overall reduction of organoid size[32]. This is in agreement with the decreased organoid size that we observed for *SURF1* mutations. It is interesting to point out that, in addition to developmental delay[9], microcephaly is often present in LS patients[72]. Collectively, the findings suggest that LS might occur more frequently than reported; the defects in early neuronal morphogenesis may prevent the normal course of brain development in some cases, leading to early termination of pregnancy.

Our study highlights two potential intervention strategies for LS caused by *SURF1* mutations, namely gene augmentation therapy (GAT) and bezafibrate treatment (Supplementary Fig. 8). There are currently no FDA-approved drugs for mitochondrial disease[18,19,54]. Using a GAT approach, we found that *SURF1* mutations caused a loss of function that could be overcome by expressing a healthy copy of *SURF1* without eliminating the diseased gene. AAV-based GAT is currently considered for various monogenic diseases of the nervous system[51]. Recently, the FDA-approved AAV-based GAT to treat children with spinal muscular atrophy[73]. In the context of LS, AAV-based GAT ameliorated the disease phenotypes in *Ndufs4* knock-out mice[74,75]. Further studies of *SURF1* GAT in living animals are needed to identify potential side effects and improve delivery strategies.

By investigating the disease pathophysiology, we found that SURF1 NPCs exhibited reduced levels of PGC1A, a master regulator of mitochondrial biogenesis[57]. Increasing mitochondrial biogenesis in SURF1 NPCs via PGC1A overexpression or via bezafibrate treatment promoted the OXPHOS shift and supported neuronal morphogenesis. Accordingly, elevating mtDNA

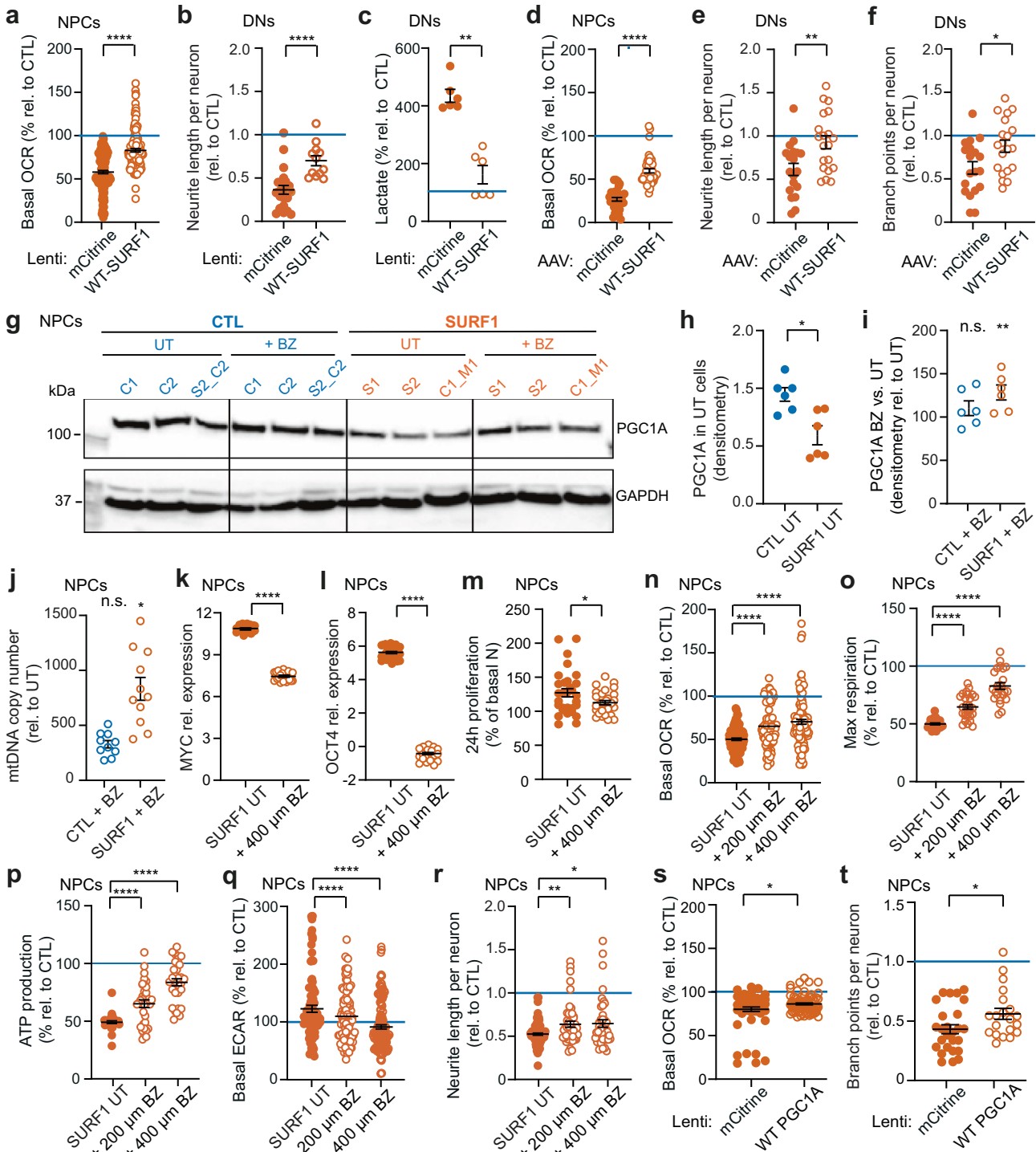

copies was beneficial in mice with pathogenic mtDNA mutations[76], and overexpression of PGC1A in *Surf1* knock-out mice improved OXPHOS activity[77]. Bezafibrate treatment has been suggested for treating neurological diseases[58,78]. Although bezafibrate did not improve mitochondrial function in *Surf1* knock-out mice[77], it was effective in delaying the accumulation of mtDNA deletions in mice[79]. A recent open-label study demonstrated the functional benefit of bezafibrate in six patients with mitochondrial cardiomyopathy[59]. The study also found an increase in serum growth factors that are associated with mitochondrial diseases, suggesting that long-term studies are needed to explore the risks and benefits of bezafibrate treatment.

Interestingly, patient S1, from whom we derived the iPSCs used in this study, was treated with bezafibrate for nine years. He beneficially responded to the treatment with improved respiratory function, reduction of dystonic episodes, and fewer metabolic crises. Possibly because of this treatment, patient S1 reached 25 years of age, which is among the oldest age reported for LS patients carrying *SURF1* mutations.

We addressed the effect of hypoxia in LS human neural cells. Hypoxia was beneficial in animal models of LS based on *Ndufs4* knock-out[52,53]. However, hypoxia exposure in SURF1 NPCs increased glycolysis and failed to promote the switch to OXPHOS and the restoration of morphogenesis. Increasing doses of glucose

**Fig. 7 SURF1 gene augmentation and bezafibrate treatment ameliorate bioenergetics and morphogenesis in patient neural cells. a** Bioenergetics of NPCs (SURF1_Mut: S1, S2) with mCitrine or WT-SURF1 (mean ± s.e.m.; n = 20 biological replicates (dots) per line over three independent experiments; ***p < 0.001; two-sided Mann–Whitney U test). **b, c** Morphogenesis and lactate in 4w DNs (SURF1_Mut: S1, S2) with mCitrine or WT-SURF1 (mean ± s.e. m.; n = 3 biological replicates (dots) per line over three independent experiments; **p = 0.0022, ****p < 0.001; two-sided Mann–Whitney U test). **d** Bioenergetics of NPCs (SURF1_Mut: S1, S2) with mCitrine or WT-SURF1 (mean ± s.e.m.; n = 5 biological replicates (dots) per line over two independent experiments; ****p < 0.0001; two-sided Mann–Whitney U test). **e, f** Morphogenesis of 4w DNs (SURF1_Mut: S1, S2) with mCitrine or WT-SURF1 (mean ± s. e.m.; n = 10 biological replicates (dots) per line over two independent experiments; *p = 0.0174, **p = 0.0061; two-sided Mann–Whitney U test). **g–i** PGC1A immunoblot and densitometry of NPCs from CTL (CTL_NoMut: C1, C2; SURF1_NoMut: S2_Corr2) and SURF1 (SURF1_Mut: S1, S2; CTL_Mut: C1_Mut1) UT or with 400 μM BZ (mean ± s.e.m.; n = 2 independent experiments; *p < 0.05 SURF1-UT vs. CTL-UT, n.s not significant CTL BZ vs. CTL-UT; **p < 0.01 SURF1 BZ vs. SURF1-UT; two-sided Mann–Whitney U test). **j** mtDNA quantification in NPCs from CTL (CTL_NoMut: C1, C2, C3; SURF1_NoMut: S2_Corr1) and SURF1 (SURF1_Mut: S2; CTL_Mut: C1_Mut1) UT or with 400 μM BZ (mean ± s.e.m.; n = 5 biological replicates (dots) per line over two independent experiments; n.s. = not significant CTL BZ vs. CTL-UT; *p < 0.05 SURF1 BZ vs. SURF1-UT; two-sided Mann–Whitney U test). **k–m** QRT-PCR (normalized to AXTB) and 24 h proliferation in NPCs (SURF1_Mut: S1, S2; CTL_Mut: C1_Mut1) UT or with BZ (mean ± s.e.m.; n = 10 replicates (dots) per line over two independent experiments; *p < 0.05, ****p < 0.0001; two-sided Mann–Whitney U test). **n–r** Bioenergetics and morphogenesis of NPCs (SURF1_Mut: S1, S2; CTL_Mut: C1_Mut1) UT or with BZ (mean ± s.e.m.; n = 15 biological replicates (dots) over three independent experiments; *p = 0.038, **p = 0.0069, ****p < 0.0001; two-sided Mann–Whitney U test). **s, t** Bioenergetics and morphogenesis of NPCs (SURF1_Mut: S1, S2; CTL_Mut: C1_Mut1) with mCitrine or PGC1A (mean ± s.e.m.; n = 10 biological replicates (dots) per line over two independent experiments; *p < 0.05; two-sided Mann–Whitney U test). Horizontal blue lines in all panels: average values CTL (CTL_NoMut: C1; SURF1_NoMut: S2_Corr1).

---

or pyruvate supplementation that has been suggested as a therapy for LS[56] also failed to boost OXPHOS in SURF1 NPCs. Further studies investigating additional metabolic manipulations and exploring the long-term consequences of hypoxia and its effects on 3D architecture should be undertaken to clarify these issues.

Our model does not provide support for the use of anti-oxidants to treat LS. It is interesting to point out that the current understanding of LS is that the disease is caused by neuronal degeneration. This interpretation had led to experimental treatment schemes centered on antioxidants to prevent the build-up of damaging free radicals[4]. However, these treatments did not show the expected positive outcomes[18,19,54]. In fact, antioxidant regiments may dampen the physiological effect of free radical signaling, thereby blunting compensatory responses[80]. Antioxidants were also found to exacerbate the susceptibility to cell death in fibroblasts from patients affected by French Canadian variant of Leigh syndrome[81]. Our findings provide a novel perspective to LS pathology by showing that the disease mechanisms may not necessarily involve a redox imbalance but rather an impairment of neuronal morphogenesis following the loss of NPC commitment.

Taken together, our data underscore the importance of metabolic programming—and particularly the metabolic shift toward OXPHOS at the level of neural progenitors—for physiological human neurogenesis, and are in agreement with recent studies demonstrating that mitochondrial metabolism is instructive for neurogenesis[82]. These findings may also help explain the presence of intellectual disability in patients with mitochondrial diseases. Our work sheds new light on the mechanisms underlying the neuronal pathology of mitochondrial disease in children and suggests implementable interventions against LS, a rare incurable pediatric disease with significant unmet medical needs.

## Methods

**Subject details.** We obtained written informed consent to use patient material from the guardians according to the Declaration of Helsinki. The study was approved by the IRB of the Charité (ethical approval EA2/131/13 and EA2/107/14). We obtained skin fibroblasts from two patients (S1 and S2) belonging to two distinct consanguineous families. Patient S1 was male with healthy parents, who were first-degree cousins from Turkey. His disease manifested with a pronounced action tremor, progressive ataxia, and episodes of hyperventilation at the age of 2 years. Cranial MRI uncovered bilateral basal ganglia necroses and T2-signal intensities in the brainstem at the *formatio reticularis* (Supplementary Fig. 1a). CSF lactate was elevated to 4.3 mmol/l (normal: <2). Muscle and skin biopsy at 2 years of age revealed low COX activity in muscles (15 mU/mg NCP; normal: 90–281) and in cultured skin fibroblasts (130 mU/U CS; normal: 680–1190), while the other respiratory chain complexes were normal. Sequencing of SURF1 (NM_003172)

showed a homozygous variant c.530T > G p.(V177G) in exon 6 that was hetero-zygous in both parents. Segregation of the mutation was verified by primer-induced restriction analysis (PIRA) (Supplementary Fig. 1b). From the age of 6 months his growth fell below the third percentile. At 18 years of age, his height was 45 cm below and his head circumference 8 cm below the third percentile. Seizures were treated successfully with Levetiracetam. From the age of 15 years, frequent metabolic crises with respiratory insufficiency required intermittent mechanical ventilation at home through a tracheostoma. His overall clinical situation considerably improved under therapy with the PPAR-agonist bezafibrate (8 mg/kg BW). The patient died at the age of 25 years from a pulmonary bacterial infection and sepsis. Patient S2 was a male from healthy consanguineous parents from Turkey. He was born with hypospadia grade II and was very hirsute from birth. Muscle weakness became evident around 2 years of age. Cranial MRI revealed the characteristic basal ganglia necrosis of LS (Supplementary Fig. 1c). At 20 months of age, biochemical investigation confirmed COX deficiency in muscle (158 mU/U CS; normal: 5202080) and in cultured fibroblasts (100 mU/U CS; normal: 342–627). Other OXPHOS complex activities were normal. Sequence analysis of SURF1 (NM_003172) detected a homozygous c.769G>A p.(G180R) variant in exon 8. Segregation of the mutation in the family was verified by PIRA (Supplementary Fig. 1d). By the age of 3.5 years, he had lost muscle force and the ability to walk, sit, and speak. Triggered by a febrile *Mycoplasma pneumoniae* infection, he developed choreiform movements and lactate levels rose to 4.5 mmol/l (normal: <2). The patient died from global respiratory and cardiac failure at 5 years of age.

Skin fibroblasts from two patients with NDUFS4 mutations (NM_002495.2) were derived at Salzburg Medical University. We obtained written informed consent to use patient material from the guardians according to the Declaration of Helsinki. The study was approved by Technical University Muenchen (ethical approval 5360/12S). Patient NDU_1 was 5-month-old male carrying the mutation c.462delA p.(K154fs) and died at 6 months of age. Patient NDU_2 was a 4 months-old female carrying the mutation c.316C > T p.(R106*); she died at 7 months of age and was previously described as P1[34].

All information regarding patient details is reported in Supplementary Data 13.

**Derivation and cultivation of iPSCs.** We reprogrammed skin fibroblasts obtained from the two SURF1 patients (S1 and S2) and two NDUFS4 patients (NDU_1 and NDU_2) using Sendai viruses (CytoTune-iPS 2.0 #A16517, Thermo Fisher Scientific, Waltham, MA, USA). Control iPSC lines were all previously generated. Dr. Heiko Lickert (Helmholtz Center Munich) kindly provided the CTL iPSC line C1, which was previously named XM001 and was generated using episomal plasmids[83]. The MDC Stem Cell Core facility kindly provided the CTL iPSC line C2, which was generated by SCVI Stanford Cardiovascular Institute using Sendai viruses and was originally named SCVI113. We previously generated the CTL iPSC line C3, which we derived using episomal plasmids and named TFBJ[31]. We purchased the human embryonic stem cell (hESC) line H1[84] from WiCell, and we used it in accordance with the German license issued to Dr. Alessandro Prigione by the Robert Koch Institute (AZ: 3.04.02/0077). We cultured all iPSCs and hESCs on Matrigel (Corning Inc., Corning, NY, USA, #734-1100)-coated plates using StemMACS iPS-Brew XF medium (Miltenyi Biotec, Bergisch Gladbach, Germany, #130-104368), supplemented with Pen/Strep (Thermo Fisher Scientific, Waltham, MA, USA, #15140122) and MycoZap (Lonza, Basel, Switzerland, #LT017-918). We routinely monitored against mycoplasma contamination using PCR. We added 10 μM ROCK inhibitor (Enzo Biochem Inc., Farmingdale, NY, USA, #ALX-270-333-M005) after splitting to promote survival. We kept PSC cultures in a humidified atmosphere of 5% CO2 at 37 °C and atmospheric oxygen. Karyotype analysis was performed by MDC Stem Cell Core Facility. Briefly, DNA was isolated using the

DNeasy blood and tissue kit (Qiagen, Venlo, Netherlands, #69504). SNP karyotyping was assessed using the Infinium OmniExpressExome-8 Kit (Illumina, San Diego, CA, USA, #200224676) and the iScan system from Illumina (Illumina, San Diego, CA, USA). CNV and SNP visualization was performed using KaryoStudio v1.4 (Illumina, San Diego, CA, USA). In Supplementary Data 13, we report all the details related to control lines, patient lines, and genome-edited lines, and their use across experiments.

**CRISPR/eCas9 genome editing**. We prepared sgRNA-Cas9 plasmids for the generation of isogenic corrected iPSC lines SURF1_NoMut (S2_Corr1, S2_Corr2, S2_Corr3) and isogenic mutant iPSC lines CTL_NoMut (C1_Mut1, C1_Mut2) following our published protocol[85]. To target the SURF1 c.769G>A mutation, we designed two sgRNAs oligo pairs targeting A>G and G>A variants using CRISPOR (http://crispor.tefor.net/) based on hg38 reference (GRCh, Genome Reference Consortium Human Reference[38]) with manual modification of G>A in sgRNA targeting the mutated SURF1 variant in S2 line. We cloned sgRNAs oligo pairs into sgRNA-Cas9 plasmid (Addgene #86986, Watertown, MA, USA) carrying enhanced specificity (eSpCas9) variant of SpCas9 with reduced off-target effects and robust on-target cleavage obtained from the eSpCas9(1.1) plasmid (Addgene, Watertown, MA, #71814)[36]. For editing by generation of double-strand breaks (DSBs) 8 nt upstream of the region of interest and repair of the DNA cleavage using single-stranded template repair pathway (SSTR) with single-stranded DNA, we designed 149 nt single-strand oligo-deoxynucleotide (ssODN) templates carrying A>G (correction) and/or G>A (introduction) variants and two silent mutations within sgRNAs binding region in close proximity to the protospacer adjacent motif (PAM) site (3 and 6 nt downstream) to prevent the templates from recurrent eCas9 cleavage in edited cells. In order to improve the DNA recombination after eCas9-induced DSBs cleavage by SSTR and ssODNs, we applied ectopic expression of human RAD52 (for SSTR upregulation) and dominant-negative sub-fragment of murine 53BP1 (dn53BP1), which may counteract the endogenous 53BP1 (for non-homologous end-joining - NHEJ downregulation)[37]. Components of the two DNA repair pathways (RAD52, dn53BP1) were kindly provided by Dr. Bruna S. Paulsen (Harvard University, Cambridge, MA, USA). We generated RAD52 and dn53BP1 expression plasmids by sub-cloning of the RAD52 and dn53BP1 PCR products into CAG expression plasmid. We carried out transient transfection of corresponding plasmids and SSTR templates in S2 and C1 lines grown in feeder-free conditions in StemMACS™ iPS-Brew XF, human culture media (Miltenyi Biotec, Bergisch Gladbach, Germany, #130-104-368) in a 6-well culture plate. One day before transfection, we dissociated the cells using Stem Pro Accutase (Thermo Fisher Scientific, Waltham, MA, USA, #A1110501) and seeded ~1 × 10⁵ cells per well of a Matrigel-coated 6-well plate as single cells or small clumps. We cultivated the cells in fresh medium containing 10 μM ROCK inhibitor overnight. We performed lipofection using the Lipofectamine 3000 Kit (Thermo Fisher Scientific, Waltham, MA, USA, #L3000015), according to the manufacturer's protocol. We diluted the plasmids to 2 mg DNA in 125 ml of Opti-MEM reduced serum medium (Gibco/ Thermo Fisher Scientific, Waltham, MA, USA, #31985062) and added the DNA-lipid complex to one well of a 6-well plate in a dropwise manner with the addition of 5 μM ROCK inhibitor to the culture medium for 24 h. We changed the medium on the following day, and then kept the cells in culture until fluorescence-activated cell sorting (FACS). We next dissociated the cells using Accutase for 5 min, washed them with PBS, and resuspended them in DPBS. We filtered the cells using Falcon polystyrene test tubes (Corning, NY, USA, #352235) and transferred them to Falcon polypropylene test tubes (Corning, NY, USA, #352063). We performed FACS-based cell sorting using BD FACSAria III at the MDC FACS Facility. We resuspended the sorted cells in recovery mTeSRTM medium (STEMCELL Technologies, Vancouver, Canada, #85850) with 1X Penicillin-Streptomycin (P/S) and ROCK inhibitor and plated them onto Matrigel-coated 6-well plates (5 K cells/ well). We transferred the growing single-cell-derived from 6-well plates to one well each of 24-well plate and maintained them until the colony grew large enough to be partially harvested for DNA isolation using Phire Animal Tissue Direct PCR Kit (Thermo Fisher Scientific, Waltham, MA, USA, #F140WH). We carried out PCR reactions using 100 ng gDNA in 50 ml with Phusion High-Fidelity DNA Taq polymerase (Thermo Fisher Scientific, Waltham, MA, USA, #F530L) according to manufacturer's instructions and annealing temperature of 61 °C. For Sanger sequencing or fragment analysis, we gel-purified the PCR products using the Wizard SV Gel and PCR Clean-Up System (Promega, Madison, WI, USA, #A9281). We amplified SURF1 gene product with S2 primers (product length 550 nt). For primer-induced restriction analysis (PIRA) the PCR product of 550 nt was cut by BbsI (NEB, Ipswich, MA, USA, #R3539) into 221 + 324 nt fragments only in the presence of the mutation c.769G>A. We submitted PCR products to LGC (https://www.lgcgroup.com, Teddington, UK) for Sanger sequencing.

Primers, sgRNA, and HDR templates sequences are reported in supplementary Supplementary Data 12.

**Whole-genome sequencing (WGS) and off-target effects analysis**. We isolated genomic DNA (gDNA) from iPSCs SURF1_Mut (S2) and SURF1_NoMut (S2_Corr1) using FlexiGene DNA isolation kit (Qiagen, Venlo, Netherlands, #51206). WGS was performed by BGI using standard procedures. In accordance with the German privacy protection laws, we are not allowed to deposit the genomic datasets on open repositories. For analysis of off-target effects of CRISPR/eCas9 genome

editing, we used the CrispRGold 1.1 algorithm (https://crisprgold.mdc-berlin.de, MDC, Berlin, Germany) for rigorous prediction of off-target sites based on applied, and manually modified (G > A) sgRNA. We next interrogated the WGS datasets of S2_Corr1 to identify potential mismatches compared to the parental line S2. The results of the analyzed top-ranked 58 sites are reported in Supplementary Data 1.

**Generation of neural progenitor cells (NPCs) and differentiated neuronal cultures (DNs)**. We obtained NPCs and DNs using a previously published protocol[39]. Briefly, we detached PSCs from Matrigel-coated plates using Accutase (1 mg/ml) and transferred the collected cells into low-attachment Petri dishes where they were kept for 2 days in Neurobasal:DMEM/F12 [1:1] (Gibco/Life Technologies, Thermo Fisher Scientific, Waltham, MA, USA, #21103049), N2 supplement [1x] (Gibco/Life Technologies, Waltham, MA, USA, #17502048), B27 supplement without vitamin A [1x] (Gibco/Life Technologies, Thermo Fisher Scientific, Waltham, MA, USA, #12587010) with the addition of Purmorphamine (PMA) [0.5 μM] (Merck Millipore, Burlington, MA, USA, #483367), CHIR 99021 [3 μM] (Cayman Chemical, Ann Arbor, MI, USA, #13122), SB-431542 [10 μM] (Selleckchem, Houston, TX, USA, #S1067). From day 2 to day 6, the media was switched to: Neurobasal:DMEM/F12 [1:1], N2 [1x], B27 without vitamin A [1x] with the addition of PMA [0.5 μM], CHIR 99021 [3 μM], ascorbic acid [150 μM] (all from Sigma-Aldrich, St. Louis, MO, USA, #A92902). On day 6, we transferred the suspended cells onto Matrigel-coated well plates using: Neurobasal:DMEM/F12 [1:1], N2 [1x], B27 without vitamin A [1x]. We maintained NPCs on this media without ROCK inhibitor and used them for experiments between passage 7 and passage 20. For DNs, we used NPCs between passage 7 and passage 13. To initiate the differentiation, we switched the media to: Neurobasal:DMEM/F12 [1:1], N2 [1x], B27 with vitamin A [1x] (Gibco/Life Technologies, Thermo Fisher Scientific, Waltham, MA, USA, #17504-044) with the addition of ascorbic acid [100 μM], FGF8 [100 ng/ml] (R&D Systems, Minneapolis, MN, USA, #4745-F8-050), PMA [0.5 μM]. After 7 days, we replaced the media condition with: Neurobasal:DMEM/ F12 [1:1], N2 [1x], B27 with vitamin A [1x] with the addition of ascorbic acid [100 μM], FGF8 [100 ng/ml], PMA [0.25 μM]. On day 9, we split the cells with Accutase and seeded on Matrigel-coated plates in Neurobasal:DMEM/F12 [1:1], N2 [1x], B27 with vitamin A [1x], with the addition of ascorbic acid [200 μM], cAMP [500 μM] (Sigma-Aldrich, St. Louis, MO, USA, #D0627), BDNF [10 ng/ml] (Miltenyi Biotec, Bergisch Gladbach, Germany, #130-093-811), GDNF [10 ng/ml] (Miltenyi Biotec, Bergisch Gladbach, Germany, #130-096-291), and TGFbeta3 [1 ng/ml] (Miltenyi Biotec, Bergisch Gladbach, Germany, #130-094-007). We added 10 μM ROCK inhibitor after each splitting to promote survival. We changed the media every 3–4 days, and then kept the differentiated cells in culture for 4, 6, and 8 weeks to reach different maturation stages. We performed magnetic-activated cell sorting (MACs) to quantify NCAM-positive cells using PSA-NCAM-PE antibody (Miltenyi Biotec, Bergisch Gladbach, Germany, #130-093-274), according to the manufacturer's instructions. For cytokine quantification in the culture media of DNs derived from H1, C1, S1, and S2. We collected the supernatants, concentrated them with Amicon 10 K (Merk Millipore Burlington, MA, USA, #C78144), and analyzed them using the Pro-inflammatory Panel I (MesoScale Discovery, Gaithersburg, MD, USA) using an electrochemiluminescent detection method. We performed all experiments with NPCs and DNs using cells grown in glucose-containing medium.

*Generation of cerebral organoids*. We generated iPSC-derived cerebral organoids according to a protocol previously described with some modifications[41,42]. Shortly, after dissociation into single-cell suspension with Accutase, we seeded 5000–10,000 cells per one well of 96-well plates in 100 μl of embryo body (EB) medium containing: DMEM/F12, KnockOut™ Serum Replacement (Thermo Fisher Scientific, Waltham, MA, USA, #10828-028), GlutaMAX™ Supplement (Thermo Fisher Scientific, Waltham, MA, USA, #35050061), MEM-NEAA (Gibco/Thermo Fisher Scientific, Waltham, MA, USA, #11140035), ESC FBS (Thermo Fisher Scientific, Waltham, MA, USA, #16141079), bFGF (PeproTech, Rocky Hill, NJ, USA, #100-18B), and 50 μM ROCK inhibitor. After 4 days, we replaced the medium with EB medium without bFGF and ROCK inhibitor. At day 6, we replaced the medium with neural induction medium (NIM: DMEM/F12, N2 supplement, Glutamax, MEM-NEAA, Heparin solution (Merck, Darmstadt, Germany, #H3149-25KU). At days 8–11, we embedded the formed organoids into Matrigel and kept them in NIM for 2 days, and in organoid differentiation medium containing 1:1 DMEM/ F12: Neurobasal, N2 supplement, B27 supplement without vitamin A, insulin, 2-ME solution (Thermo Fisher Scientific, Waltham, MA, USA, #31350-010), Glutamax, MEM-NEAA without retinoic acid (RA) (Sigma-Aldrich, St. Louis, MO, USA, #R2625-50MG), CHIR 99021 for another 4 days. Next, we transferred the organoids to ultra-low-attachment 6-well plates (Corning, NY, USA, #3261) and cultured them on an orbital shaker (80 rpm) in organoid maturation medium containing 1:1 DMEM/F12: Neurobasal, N2 supplement, B27 supplement with vitamin A, insulin, 2-ME solution, Glutamax supplement, MEM-NEAA, Sodium Bicarbonate, Vitamin C solution, 1× chemically defined lipid concentrate (Thermo Fisher Scientific, Waltham, MA, USA, #11905031), 0.4 mM ascorbic acid, BDNF, GDNF, cAMP, 20 ng/ml Matrigel, and HEPES. We conducted all experiments using organoids grown in glucose-containing maturation medium. We analyzed organoid sizes by measuring the area using ImageJ software. For immunostaining, each human cerebral organoid was fixed in 4% paraformaldehyde (PFA) (EMS,

Thermo Fisher Scientific, Waltham, MA, USA, #50980487) overnight at 4 °C, dehydrated by 40% sucrose in PBS, and embedded in Tissue-Tek® O.C.T.TM compound (Sakura® Finetek, Netherlands, #25608-930). We cut 12-μm sections and mounted them onto slides (Thermo Fisher Scientific, Waltham, MA, USA, #6776214).

**Droplet-based single-cell RNA-sequencing (Drop-seq).** We carried out single-cell RNA-sequencing[86] for 4w DNs derived from CTL (CTL_NoMut: C1; SURF1_NoMut: S2_Corr1) and SURF1 (SURF1_Mut: S2), and for D90 brain organoids derived from CTL (CTL_NoMut: C1; SURF1_NoMut: S2_Corr1) and SURF1 (SURF1_Mut: S2). All samples were assessed in two independent experiments. For brain organoids, we pooled 3–8 organoids per line per experiment. We processed raw paired-end scRNA-seq data to generate a digital gene expression (DGE) matrix using Drop-seq tools v. 2.3.0 with default parameters[87]. We centrifuged methanol-fixed cells at 3000–5000 × g for 5 min, we rehydrated them in 1 ml PBS plus 0.01% BSA supplemented (Sigma-Aldrich, St. Louis, MO, USA, #A9576-50ML) with RNAse inhibitors (1 unit/μl RiboLock, Thermo Fisher Scientific, Waltham, MA, USA, #EO0381). We pelleted them and resuspended them in 0.5 ml PBS plus 0.01% BSA in the presence of RNAse inhibitors. We manually counted the cells by means of a hemocytometer and we diluted them to a suspension of typically ~200 cells/μl in PBS plus 0.01% BSA. We encapsulated the cells together with Barcoded microparticles (Barcoded Beads SeqB #Macosko-2011-10; ChemGenes Corp., Wilmington, MA, USA) using a self-built Drop-seq set up (Online-Drop-seq-Protocol-v.3.1: http://mccarrolllab.com/dropseq/) as previously described[88]. Droplets were broken immediately after collection. We reverse-transcribed and exonuclease-treated the barcoded beads with captured transcriptomes. We amplified the first-strand cDNA by equally distributing beads from one run to 24 PCR reactions (50 μl volume; 4 + 9 to 11 cycles). We pooled 20 μl fractions of each PCR reaction (total = 480 μl), then we double-purified with 0.6× volumes of AMPure XP beads (Beckman Coulter, Brea, CA, USA). We assessed amplified cDNA libraries and quantified using a BioAnalyzer High Sensitivity Chip (Agilent, Santa Clara, CA, USA) and the Qubit dsDNA HS Assay system (Thermo Fisher Scientific, Waltham, MA, USA, #Q32851). We fragmented 600 pg of each cDNA library, amplified (12 cycles) them, and indexed them for sequencing with the Nextera XT v2 DNA sample preparation kit (Illumina, San Diego, CA, USA, #131-1024) using custom primers enabling 3'-targeted amplification. We purified the libraries with AMPure XP Beads, quantified them, and sequenced them on Illumina NextSeq 500 sequencers (Illumina Inc., San Diego, CA, USA), NextSeq 500/550 High Output v2 kit (75 cycles) (Illumina, San Diego, CA, USA, #2002496) in paired-end mode. We processed raw paired-end scRNA-seq data using Drop-seq tools v. 2.3.0 with default parameters (https://github.com/broadinstitute/Drop-seq/releases/tag/v2.3.0) to generate the DGE matrices. We performed alignment to the hg19 reference genome using STAR v. 2.6.0[89]. We achieved unique mapping for around 80% of the reads in DNs, and 60–70% of the reads in organoids. We discarded non-uniquely mapped reads. To distinguish between beads that captured cellular transcriptomes from those that captured ambient RNA, we sorted barcodes by decreasing number of reads and picked the inflection point ('knee') of the cumulative fraction of reads plot. We selected the top 1000 barcodes for DNs, the top 8000 for organoids from S2 and S2_Corr1, and the top 3500 for organoids from C1. We used Seurat v. 3.1.0 for downstream computational analyses[90]. To remove damaged cells, we extracted the percentage of mitochondrial reads and the count of captured transcripts (nCount_RNA) and removed all barcodes with <400 nCount_RNA or high-percentage of mitochondrial reads (>40% in 2D cultures and >20% in 3D cultures). We removed barcodes with extremely low mitochondrial reads (<0.8%) to exclude nuclei. In order to exclude potential doublets, we also excluded cells with very high nUMI (>5000). For each cell, UMI counts per gene were normalized and scaled. We performed clustering considering only the top 1000 highly variable genes, as identified by the function "FindVariableGenes". Variable genes were then used to perform principal component (PC) analysis. We selected the PCs to be used for downstream analyses by evaluating the "PCElbowPlot" and the "JackStrawPlot". We used the first 20 PCs for DNs, and the first 30 PCs for organoids. We identified clusters using the function "FindClusters", which exploits a SNN modularity optimization clustering algorithm (at Resolution=1 for DNs and Resolution=0.6 for organoids). We visualized clusters using the uniform manifold approximation and projection (UMAP) dimensionality reduction[44]. We used the manual inspection of marker genes determined using the "FindAllMarkers" function for cluster identification. This function determines which genes, that are expressed in at least three cells, are enriched in every clustering using a Wilcoxon rank-sum test. In DNs, we removed the cluster containing high level of nuclear marker (MALAT1); in organoids, we removed the cluster characterized by high expression of hemoglobin genes (HBD and HBB) and interferon-response genes (e.g., IFIT1). After cluster removal, we performed again normalization, dimensionality reduction, and clustering. Finally, we assigned each cell to a cell cycle phase (G1, 2, G2M) using the "CellCycleScore" function and a published gene set[91].

**Bulk RNA-sequencing.** We carried out ribo-zero total RNA-sequencing for 4w DNs and 8w DNs derived from CTL (SURF1_NoMut: S2_Corr1) and SURF1 (SURF1_Mut: S2), and for D90 brain organoids derived from CTL (SURF1_NoMut: S2_Corr1) and SURF1 (SURF1_Mut: S2). All samples were assessed in three

independent experiments. We used 500 ng of total RNA, where rRNA was depleted using RNase H-based protocol. We mixed total RNA with 1 μg of a DNA oligo-nucleotide pool comprising 50-nt long oligonucleotide mix covering the reverse complement of the entire length of each human rRNA (28S rRNA, 18S rRNA, 16S rRNA, 5.8S rRNA, 5S rRNA, 12S rRNA), incubated with 1U of RNase H (Hybridase Thermostable RNase H, #H39500, Epicenter Technologies Pvt. Ltd, Thane, India), purified using RNA Cleanup XP beads (Agencourt #001298v001, Beverly, MA, USA), DNase treated using TURBO DNase rigorous treatment protocol, and purified again with RNA Cleanup XP beads. We fragmented the rRNA-depleted RNA samples and processed them into strand-specific cDNA libraries using TruSeq Stranded Total LT Sample Prep Kit (Illumina, San Diego, CA, USA, #20020596) and then sequenced them on NextSeq 500, High Output Kit, 2 × 76 cycles (Illumina, San Diego, CA, USA, #2002496). We carried out polyA mRNA-sequencing for 4w and 8w DNs derived from H1, C1, S1, and S2 (all in biological triplicates). mRNA-seq was performed by BGI using an oligo dT selection (mRNA enrichment) strategy with oligo dT beads to select mRNA with polyA tail using BGISEQ-500 with DNB seq technology. We mapped all raw sequencing reads to the human genome (GRCh38 assembly) using STAR (version 2.6.0c) aligner[92]. We used the default settings, except out-FilterMismatchNoverLmax, which was set to 0.05. We counted reads using the htseq-count tool, version 0.9.1[93], with gene annotation from GENCODE release 27[94]. For mRNA-sequencing, we analyzed 8w DNs derived from hESCs (H1), CTL (CTL_NoMut: C1) and SURF1 (SURF1_Mut: S1, S2) using the same DESeq2 parameters. All samples were assessed using three biological replicates out of three independent experiments. We summed up read counts for genes expressed in H1, C1, S1, and S2 across the triplicates. S1 and S2 were treated as disease replicates and compared to H1 and C1. For total RNA-sequencing, we compared S2 to S2_Corr1. We performed differential gene expression analysis using the DESeq2 (version 1.20.00) R package[95], with default statistical analysis: two-tailed Wald test with multiple comparisons correction according to Benjamini–Hochberg procedure. All genes with a false discovery rate (FDR) < 0.05 were considered differentially expressed. We performed functional enrichment analysis using the gProfileR R package version 0.6.6, with default settings[96]. All expressed genes were used as background.

**Proteomics analysis.** We carried out label-free quantification (LFQ) proteomics with sample preparation according to a published protocol with minor modifications[97]. We used biological triplicates of 8w DNs derived from CTL (SURF1_NoMut: S2_Corr1) and SURF1 (SURF1_Mut: S2). We lysed samples under denaturing conditions in a buffer containing 3 M guanidinium chloride (GdmCl), 5 mM tris(2-carboxyethyl)phosphine, 20 mM chloroacetamide, and 50 mM Tris-HCl pH 8.5. Lysates were denatured at 95 °C for 10 min shaking at 1000 rpm in a thermal shaker and sonicated in a water bath for 10 min. We used a small aliquot of cell lysate for the BCA assay to quantify the protein concentration. We diluted the lysates (100 μg proteins) with a dilution buffer containing 10% acetonitrile and 25 mM Tris-HCl, pH 8.0, to reach a 1 M GdmCl concentration. We digested proteins with 1 μg LysC (MS grade, Roche, Basel, Switzerland) shaking at 700 rpm at 37 °C for 2 h. We diluted the digestion mixture with the same dilution buffer to reach 0.5 M GdmCl. We added 1 μg trypsin (MS grade, Roche) and incubated the digestion mixture at 37 °C overnight in a thermal shaker at 700 rpm. We used solid-phase extraction (SPE) disc cartridges (C18-SD, Waters, Milford, MA, USA) for peptide desalting, according to the manufacturer's instructions. We reconstituted desalted peptides in 0.1% formic acid in water and further separated them into four fractions by strong cation exchange chromatography (SCX, 3 M Purification, Meriden, CT, USA). We dried eluates in a SpeedVac, dissolved them in 20 μl 5% acetonitrile and 2% formic acid in water, briefly vortexed them, and sonicated them in a water bath for 30 s prior injection to nano-LC-MS. LC-MS/MS was carried out by nanoflow reverse-phase liquid chromatography (Dionex Ultimate 3000, Thermo Fisher Scientific, Waltham, MA, USA) coupled online to a Q-Exactive HF Orbitrap mass spectrometer (Thermo Fisher Scientific, Waltham, MA, USA). Briefly, we performed the LC separation using a PicoFrit analytical column (75 μm ID × 55-cm long, 15 μm Tip ID; New Objectives, Woburn, MA, USA) in-house packed with 2.1-μm C18 resin (Reprosil-AQ Pur, Dr. Maisch, Ammerbuch, Germany). We eluted peptides using a gradient from 3.8 to 40% solvent B in solvent A over 120 min at 266 nL per minute flow rate. Solvent A was 0.1% formic acid and solvent B was 79.9% acetonitrile, 20% $H_2O$, 0.1% formic acid. Nanoelectrospray was generated by applying 3.5 kV. A cycle of one full Fourier transformation scan mass spectrum (300−1750 m/z, resolution of 60,000 at m/z 200, AGC target 1e6) was followed by 12 data-dependent MS/MS scans (resolution of 30,000, AGC target 5e5) with a normalized collision energy of 25 eV. In order to avoid repeated sequencing of the same peptides, we used a dynamic exclusion window of 30 s. In addition, we sequenced only peptide charge states between two to eight. We processed raw MS data with MaxQuant software (v1.6.0.1) and searched against the human proteome database UniProtKB with 21,074 entries, released on 12/2018. Parameters of MaxQuant database searching were false-discovery rate (FDR) of 0.01 for proteins and peptides, a minimum peptide length of 7 amino acids, a mass tolerance of 4.5 ppm for precursor, and 20 ppm for fragment ions. We used the function "match between runs". A maximum of two missed cleavages was allowed for the tryptic digest. We set cysteine carbamido-methylation as fixed modification, while N-terminal acetylation and methionine

oxidation were set as variable modifications. We strictly excluded from further analysis any contaminants, as well as proteins identified by site modification and proteins derived from the reversed part of the decoy database. We report the MaxQuant processed output files, peptide and protein identification, accession numbers, % sequence coverage of the protein, $q$-values, and LFQ intensities in Supplementary Data 6. We performed the correlation analysis of biological replicates and the calculation of significantly different metabolites and proteins using Perseus (v1.6.5.0). LFQ intensities, originating from at least two different peptides per protein group, were transformed by log2. We employed only protein groups with valid values within compared experiments. We carried out statistical analysis by a two-sample two-tailed $t$-test with Benjamini–Hochberg (BH, FDR of 0.05) correction for multiple testing. Significantly regulated metabolites and proteins between patients and controls were indicated by a plus sign in Supplementary Data 6. For comprehensive proteome data analyses, we applied gene set enrichment analysis (GSEA, v2.2.3)[98] in order to see, if a priori defined sets of proteins show statistically significant, concordant differences between mutations and controls. For GSEA analysis, we used all proteins with ratios calculated by Perseus. We applied GSEA standard settings, except that the minimum size exclusion was set to 5 and Reactome v5.2 and KEGG v5.2 were used as gene set databases. The cutoff for significantly regulated pathways was set to be ≤0.05 $p$-value and ≤0.05 FDR. We report the results of the GSEA analysis in Supplementary Data 7. The mass spectrometry data is deposited in the ProteomeXchange Consortium (http://proteomecentral.proteomexchange.org) via the PRIDE partner repository[99].

**Metabolomics analysis and multi-omics integration.** We carried out metabolite extraction and profiling by targeted LC-MS as reported previously[100]. We harvested biological triplicates of 8w DNs derived from CTL (SURF1_NoMut: S2_Corr1) and SURF1 (SURF1_Mut: S2). We aspirated the culture medium, quickly rinsed the cells twice with ice-chilled 1x PBS, pelleted the cells, and shock-freeze them in liquid nitrogen. We extracted the metabolites with methyl tert-butyl-ether (MTBE), methanol, and water. The remaining protein pellet was used in BCA protein assay for normalization among samples. We aliquoted the extracts equally into three tubes for later reconstitution in water, acetonitrile, and 50% methanol in acetonitrile, respectively. We added to each sample an internal standard mixture containing chloramphenicol, C13-labeled L-glutamine, L-arginine, L-proline, L-valine, and uracil (10 μM final concentration). A SpeedVac was used to dry the aliquots. We dissolved dry residuals in three different solvents: (1) 100 μL 50% acetonitrile in MeOH with 0.1% formic acid, (2) 100 μL MeOH with 0.1% formic acid for analysis by HILIC column, or (3) 100 μL water, 0.1% formic acid for C18 column mode. We transferred the supernatants to micro-volume inserts. We injected 20 μL per run for subsequent LC-MS analysis. Over 400 metabolites were selected to cover most of the important metabolic pathways in mammals. Since metabolites are very diverse in their chemical properties, we used two different LC columns for metabolite separation: Reprosil-PUR C18-AQ (1.9 μm, 120 Å, 150 × 2 mm ID; Dr. Maisch, Ammerbuch, Germany) and zicHILIC (3.5 μm, 100 Å, 150 × 2.1 mm ID; Merck, Darmstadt, Germany). We used the settings of the LC-MS instrument, 1290 series UHPLC (Agilent, Santa Clara, CA, USA) online coupled to a QTrap 6500 (Sciex, Foster City, CA, USA) as reported previously[101]. The buffer conditions were A1, 10 mM ammonium acetate, pH 3.5 (adjusted with acetic acid); B1, 99.9% acetonitrile with 0.1% formic acid; A2, 10 mM ammonium acetate, pH 7.5 (adjusted with ammonia solution); B2, 99.9% methanol with 0.1% formic acid. We prepared all buffers in LC-MS grade water and organic solvents. We performed peak integration with MultiQuantTM software v.2.1.1 (Sciex, Foster City, CA, USA) and reviewed it manually. We normalized peak intensities, first against the internal standards, and subsequently against protein abundances obtained from the BCA assay. We used the first transition of each metabolite for relative quantification between samples and controls. We carried out statistical analysis with Perseus (https://www.nature.com/articles/nmeth.3901). We employed a two-sample two-tailed t-test with Benjamini–Hochberg (BH, FDR of 0.05) correction for multiple testing without $q$-value cutoff. All values above the dashed line in Fig. 4e are significant by this measure. In order to also address the biological meaning of large ratios between sample means, we calculated a second and independent significance cutoff indicated by a solid line in Fig. 4e. This cutoff was based on an FDR of 0.05, while the minimal fold change s0 was set to 0.1. In this way, if a metabolite gives a very good $p$-value but its fold change is below s0, it will not be considered as significant. Using this approach, we could take into account both the $p$-value (0.05) and the difference between the means. We provide the list of all metabolites including MRM ion ratios, KEGG and HMDB metabolite identifiers, and statistical values in Supplementary Data 8. These data were obtained using a previously reported LC-MS method containing the list of metabolites, transitions, and retention times[102]. We deposited all original LC-MS generated QTrap wiff files as well as MultiQuantTM processed peak integration q.session on the peptideatlas repository (http://www.peptideatlas.org). We performed the integration of total RNA-sequencing, proteomics, and metabolomics of 8w DNs derived from S2 and S2_Corr1 using xMWAS, as previously described[45].

**PCR analyses and nanostring.** We performed gene expression analysis by quantitative real-time RT-PCR (qPCR) using SYBR Green PCR Master Mix (Thermo Fisher Scientific, Waltham, MA, USA, #4309155) and the ViiATM 7 Real-Time PCR System (Applied Biosystems, Waltham, MA, USA). For each target

gene, we measured cDNA samples and negative controls in triplicates using ABI PRISMTM 384-Well Clear Optical Reaction Plate (Applied Biosystems/Thermo Fisher Scientific, Waltham, MA, USA, #4309849). We calculated the relative transcript levels of each gene based on the $2^{-\Delta\Delta CT}$ method. We normalized the data to the housekeeping gene AXTB and presented the results as mean LOG2 ratios in relation to control cell lines. For primer-induced restriction analysis (PIRA) of S1, the PCR product of 141 bp was cut by $SmaI$ (NEB, Ipswich, MA, USA, #R0141S) into 23 + 118 bp in the presence of c.530T>G mutation. For PIRA of S2, the PCR product of 437 bp was cut by $AvaII$ (NEB, Ipswich, MA, USA, #R0153S) into 292 + 145 bp fragments only in the presence of the mutation c.769G > A. For PIRA of S2, the PCR product of 550 bp was cut by $BbsI$ (NEB, Ipswich, MA, USA, #R3539) into 221 + 324 bp fragments only in the presence of the mutation c.769G>A. For mtDNA copy number quantification, we isolated genomic DNA (gDNA) using the NucleoSpin Tissue kit (Macherey-Nagel, Düren, Germany, #740952.50). We amplified the purified gDNA using the primer pairs for $MT-ND1$ (mtDNA-encoded gene of CI) and $NDUFV1$ (single-copy nuclear-encoded gene of CI) and visualized with a gel imager (VWR). We used a MicroAmp Optical 96-Well reaction plate (Applied Biosystems/Thermo Fisher Scientific, Waltham, MA, USA, #4316813) based on the SYBR Green protocol on an ABI Prism 7000 sequence detection system. We used the ABI 7000 system SDS software to analyze the amplification curves[103]. As $MT-ND1$ is encoded by the mtDNA and $NDUFV1$ is encoded by the nuclear DNA, the ratio MT-ND1/NDUFV1*2 (mtDNA/gDNA) equals mtDNA copy number per cell[104], providing an indirect measure of the abundance of mitochondria per cell. All primer sequences are reported in Supplementary Data 12. We performed Nanostring-based differential expression analyses of mRNA expression using custom-designed 72-plex Nano-string nCounterTM probes panel. The mRNA transcript quantification analysis, including sample preparation protocol, hybridization, and detection was performed according to manufacturer's recommendations.

**Immunostaining.** We fixed cells grown on Matrigel-coated coverslips with 4% PFA for 20 min at room temperature (RT) and washed two times with PBS. For permeabilization, we incubated the fixed cells with a blocking solution containing 10% normal donkey serum (DNS) (Merck Millipore, Burlington, MA, USA, #S30-100ML) and 1% Triton X-100 (Sigma-Aldrich, St. Louis, MO, USA, #T8787) in PBS with 0.05% Tween 20 (Sigma-Aldrich, St. Louis, MO, USA, #P9416) (PBS-T) for 1 h at RT. We diluted primary antibodies in blocking solution and incubated them overnight at 4 °C on a shaker. Primary antibodies used were as follows: PAX6 (BioLegend, San Diego, CA, USA, #901301; 1:200), SOX2 (Santa Cruz, Dallas, TX, USA, #sc-17320; 1:100), TUJ1 (Sigma-Aldrich, St. Louis, MO, USA, #T8578; 1:3000), LIN28 (ProteinTech Europe, Manchester, UK, #11724; 1:300), TRA-1-60 (Millipore, Burlington, MA, USA, MAB4360; 1:200), MAP2 (Synaptic System, Göttingen, Germany, #188 004; 1:100), GFAP (Synaptic Systems, Göttingen, Germany, #173004; 1:500), NANOG (R&D Systems, Minneapolis, MN, USA, #AF1997; 1:200), smooth muscle actin (SMA) (DakoCytomation, Glostrup, Denmark, #M0851; 1:200), SOX17 (R&D Systems, Minneapolis, MN, USA, #AF1924; 1:50), TH (Millipore, Burlington, MA, USA, #AB152; 1:300), FOXA2 (Seven Hills Bioreagents, Cincinnati, OH, USA, # WRAB 1200; 1:100), S100ß (Abcam, Cambridge, UK, #14849; 1:500), SYP (Sigma-Aldrich, St. Louis, MO, USA, #SVP-38; 1:500), SYN (Invitrogen/Thermo Fisher Scientific, Waltham, MA, USA, #A-6442, 1:100), VAMP2 (Synaptic Systems, Göttingen, Germany, #104211; 1:500), NESTIN (Millipore, Burlington, MA, USA, #MAB5326; 1:200), NURR1 (Sigma-Aldrich, St. Louis, MO, USA, N4663; 1:500), pVIM (MBL Life Science, Woburn, MA, USA, #D095-3; 1:1000). Corresponding secondary antibodies (all Alexa Fluor, 1:2000, Life Technologies/Thermo Fisher Scientific, Waltham, MA, USA) were diluted in blocking solution for 1 h at RT on a shaker. Counterstaining of nuclei was carried out using 1:10,000 Hoechst (Invitrogen/Thermo Fisher Scientific, Waltham, MA, USA, #H3570). We acquired the images of 2D cultures using the confocal microscope LSM510 Meta (Zeiss, Jena, Germany) in combination with the Axio-Vision V4.6.3.0 software (Zeiss, Jena, Germany) and further processed with AxioVision software and ImageJ. We acquired images of 3D organoids using a Keyence bz-x710 (Osaka, Japan) microscope.

**Immunoblotting.** We lysed NPCs in RIPA buffer (150 mM NaCl, 5 mM EDTA, 50 mM Tris, 1% NP-40, 0.5% sodium deoxycholate and 0.1% SDS) in the presence of protease and phosphatase inhibitors. We incubated the samples on ice for 30 min and centrifuged at $14,000 \times g$ for 20 min at 4 °C. We determined protein concentration using the PierceTM BCA assay (Thermo Fisher Scientific, Waltham, MA, USA, #23225). For MT-CO2 immunoblot, we loaded 120 μg proteins on NuPAGE Novex 4–12% Bis-Tris precast SDS-PAGE gels (Invitrogen/Thermo Fisher Scientific, Waltham, MA, USA, #NP0326BOX). For PGC1A immunoblot, we loaded 40 μg of proteins in a 4–12% SDS-PAGE. We transferred the gels onto Immobilin-FL PVDF membranes (Merck Millipore, Burlington, MA, USA, #IPFL00010). We used the following primary antibodies: MT-CO2 (Abcam, Cambridge, UK, ab110258; 1:1500), beta-actin (ACTB; Sigma-Aldrich, St. Louis, MO, USA, #A5316; 1:4000), PGC1A (Novus Biological, Centennial, CO, USA, #NBP1-04676; 1:1000), and GAPDH (Immunological Sciences, Rome, Italy, #MAB-10578; 1:1000). For MT-CO2 immunoblots, we measured chemiluminescence in a Fujifilm LAS-3000 (Fujifilm, Tokyo, Japan) after the addition of Pierce ECL Western Blotting Substrate (Thermo Fisher Scientific, Waltham, MA, USA).

For PGC1A immunoblots, we acquired the images using the ChemiDoc Imaging System apparatus (Bio-Rad Laboratories, Milan, Italy) and analyzed them with Image LabTM software, version 6.0.1 for Windows (Bio-Rad Laboratories, Milan, Italy). Blue-native gel electrophoresis (BNGE) was performed as described[105]. We solubilized mitoplasts from NPCs derived from C1 and C1_Mut1 using n-dodecyl-b-d-maltoside (DDM) at 1% final concentration. We separated samples by electrophoresis using precast NativePAGE 4–16% Bis-Tris gels (Invitrogen/Thermo Fisher Scientific, Waltham, MA, USA, #BN1001BOX) followed by a second, denaturing electrophoresis using precast NuPAGETM 4–12% Bis-Tris Protein Gels, 2D-well (Invitrogen/Thermo Fisher Scientific, Waltham, MA, USA, #NP0326BOX). We transferred the samples onto a nitrocellulose membrane and immunoblotted using the OXPHOS rodent antibody cocktail (containing a cocktail of five antibodies: SDHB, UQCRC2, MT-CO1, ATP5A, and NDUFB8) (Abcam, Cambridge, UK, #ab110413), COX4I1 antibody (Invitrogen/Thermo Fisher Scientific, Waltham, MA, USA, #A21347), and SDHA antibody (Abcam, Cambridge, UK, #ab14715). We performed chemiluminescence-based immunostaining (Amersham ECL Western Blotting Detection Kit, #RPN2108, Cytiva, Marlborough, MA, USA). We acquired the images with ChemiDoc Imaging System apparatus (Bio-Rad Laboratories, Milan, Italy) and analyzed them with Image LabTM software, version 6.0.1 for Windows (Bio-Rad Laboratories, Milan, Italy).

**Detection of in situ COX enzyme activity.** We assessed the activities of cytochrome c oxidase (COX) for CIV and of succinate dehydrogenase (SDH) for CII using an in situ enzymatic colorimetric-based assay on NPCs derived from CTL (CTL_NoMut: C1, C2, C3; SURF1_NoMut: S2_Corr1) and SURF1 (SURF1_Mut: S1, S2; CTL_Mut: C1_Corr1). We performed enzyme histochemical staining using standard procedures[106]. After gentle centrifugation, we transferred the samples on Tissue Tek® on a cork plate and shock frozen them in isopentane precooled in liquid nitrogen. We generated 10-μm-thick cryosections and stained them with SDH (Abcam, Cambridge, UK, #Ab14715), COX and hematoxylin-eosin (HE) for visualization of the cells. For quantification of the enzyme activity, we visualized stained cryosections with Leica DMI6000 microscope (×10) (Leica Microsystems, Wetzlar, Germany) and took pictures using a Moticam 2500 (5.0 M Pixel) (Motic, Speed Fair Co., Ltd, Hong Kong, China) with the software Motic Images Plus 2.0. We analyzed the images using the software MBF ImageJ bundle (formerly WCIF ImageJ). The final average gray value per region of interest represented the intensity of the brown DAB deposits in the mitochondrial matrix and thereby the COX activity of different samples.

**Bioenergetic profiling.** We performed live-cell assessment of cellular bioenergetics using Seahorse XF96 extracellular flux analyzer (Agilent Technologies, Santa Clara, CA, USA), as described previously[31]. All experiments were conducted using cells grown in glucose-containing medium. Briefly, we plated 20,000 cells onto each Matrigel-coated well of XF96-well plates (Agilent Technologies, Santa Clara, CA, USA, #101085-004). We maintained NPCs in the plates for 2 days and DNs cultures for 4 or 8 weeks. Brain organoids were dissociated using Worthington papain solution and 20,000 cells were plated onto each Matrigel-coated well of XF96-well plates. On the assay day, we incubated all the cells at 37 °C 5% CO₂ for 60 min to allow media temperature and pH to reach equilibrium. We measured simultaneously mitochondrial respiration (oxygen consumption rate, OCR) and anaerobic glycolysis (extracellular acidification rate, ECAR) using the sequential introduction of oligomycin (Sigma-Aldrich, St. Louis, MO, USA, #579-13-5), FCCP (Sigma-Aldrich, St. Louis, MO, USA, #C2920-10MG), and then rotenone (Sigma-Aldrich, St. Louis, MO, USA, #83-794) plus antimycin A (Sigma-Aldrich, St. Louis, MO, USA, #1397-94-0) (all products at 1 μM). The addition of these drugs allowed us to calculate: basal OCR level, basal ECAR level, maximal respiration, and ATP production, as described in detail before[107]. We normalized the values based on the DNA content in each well of the plate quantified using CyQUANT kit (Molecular Probes/Thermo Fisher Scientific, Waltham, MA, USA, #C7026). We collected the supernatants from the XF96-well plates and used them to quantify the lactate amount with a Lactate Fluorometric Assay Kit (BioVision, Milpitas, CA, USA, #7K607-100).

**High-content analysis (HCA) of neuronal morphogenesis.** We employed HCA to quantify the neuronal branching outgrowth and complexity using CellInsight CX7 microscope (Thermo Fisher Scientific, Waltham, MA, USA). Briefly, we split NPCs or DNs at 4 or 8 weeks of differentiation using Accutase and seeded them at a density of 10,000 cells/well on Matrigel-coated 96-well plates with black-wall and clear-bottom (Corning, Corning, NY, USA, #353219). We then stained the cells with TUJ1 antibody (Sigma-Aldrich, St. Louis, MO, USA, #T8578; 1:3000) using 4% PFA (EMS, Thermo Fisher Scientific, Waltham, MA, USA, #50980487) for 20 min at RT and washed two times with PBS. For permeabilization, we incubated the fixed cells with a blocking solution containing 10% normal donkey serum (DNS) and 1% Triton X-100 (Sigma-Aldrich, St. Louis, MO, USA, #T8787) in PBS with 0.05% Tween 20 (Sigma-Aldrich, St. Louis, MO, USA, #P9416) (PBS-T) for 1 h at RT. We diluted primary antibodies in blocking solution, incubated them overnight at 4 °C on a shaker, and performed counterstaining with Hoechst. The morphogenesis of TUJ1-positive cells within NPCs or DN cultures were quantified using the "Cellomics Neuronal Profiling v4 BioApplication" (CellInsight CX7, High

Content Platform, Thermo Fisher Scientific, Waltham, MA, USA). We used a similar HCA-based approach to quantify the number of TUJ1-positive neurons and TH-positive neurons within DNs.

**Mitochondrial morphology and functionality.** For visualization of mitochondrial morphology in NPCs derived from C1, C2, S1, S2, S2_Corr1, _Corr2, we used transmission electron microscope (TEM). Ultrastructural analysis was performed after fixation of the cells in 2.5% glutaraldehyde for 48 h at 4 °C, post-fixation in 1% osmium tetroxide, and sample embedding in Araldite. Semi-thin sections were used to identify viable and characteristic cells and respective ultrathin sections were stained with uranyl acetate and lead citrate. A P902 electron microscope (Zeiss, Oberkochem, Germany) was used to analyze the specimens. For mitochondrial membrane potential (MMP) quantification, we used an HCA-based live-cell detection assay that we previously described[31]. Briefly, we split NDUFS4 NPCs and CTL NPCs with Accutase isolation and seeded them on a black-wall, clear-bottom plate coated with Matrigel at a density of 40,000 or 80,000 cells/well on 96-well plates (Corning, Corning, NY, USA, #353219) and incubated in NPC medium overnight at 37 °C, 5% CO₂. On the day of the assay, we live-stained NPCs with 10 nM TMRM (Molecular Probes, Life Technologies, Waltham, MA, USA, #T668), a potentiometric dye that accumulates in active mitochondria, for 30 min at 37 °C and 5% CO₂. We performed a control staining in parallel by exposing cells to 1 mM FCCP (Sigma-Aldrich, St. Louis, MO, USA, #C2920-10MG) and 1 mM antimycin A (Sigma-Aldrich, St. Louis, MO, USA, #1397-94-0) to cause complete mitochondrial depolarization. We then washed all cells with PBS and stained them with 1:10,000 Hoechst (33342, Invitrogen/Thermo Fisher Scientific, Waltham, MA, USA, #H3570) diluted in phenol red-free-DMEM (Gibco/Thermo Fisher Scientific, Waltham, MA, USA, #A1443001) for 10 min at RT. After additional PBS washes, we kept the cells in phenol red-free-DMEM for the duration of the assay. We imaged and analyzed the live-cells using the CellInsight CX7 microscope (Thermo Fisher Scientific, Waltham, MA, USA). For quantification of mitochondrial ROS, we seeded 250,000 NPCs per well a 24-well plate and incubated at 37 °C and 5% CO₂ until they reached 70–90% confluency. We used 3 μM antimycin A for 1 h to induce ROS production by blocking CIII. After one washing step with DPBS, we incubated the cells with 3.5 μM superoxide indicator MitoSOX (Invitrogen/Thermo Fisher Scientific, Waltham, MA, USA, #M36008) for 20–30 min at 37 °C and 5% CO₂. Within mitochondria, MitoSOX is oxidized by ROS resulting in red fluorescence (510/580 nm). We harvested the cells using Accutase and pelleted them in DPBS containing DAPI (1 μg/μL). We then carried out fluorescence-activated cell sorting (FACS) analysis to quantify the MitoSOX and DAPI signals using the LSRFortessa cell analyzer (BD, Franklin Lakes, NJ, USA). To assess proliferation, we seeded NPCs at a density of 5000, 20,000, and 40,000 cells/well onto Matrigel-coated black-wall, clear-bottom plates (Corning, NY, USA, #353219). We used two plates for samples that we fixed at 24 h distance. After fixation for 20 min in 4% PFA and 8.1 μM Hoechst, we washed the plates with PBS and analyzed their fluorescence intensity with a Tecan plate reader (Infinite M200, Tecan, Zürich, Switzerland).

**Electrophysiology.** To analyze passive and active membrane properties and spiking and synaptic activity, we carried out whole-cell patch-clamp recordings on DNs at 4, 6, and 8 weeks of differentiation. We visualized the cells at RT under phase contrast optics on an upright microscope (Axioskop, Zeiss, Germany) using a ×63/0.95 water immersion objective. We performed recordings with a patch-clamp amplifier (EPC-9, HEKA Electronics, Lambrecht, Germany). We filled recording pipettes with an intracellular solution containing (in mM): 4 NaCl, 120 KCl, 5 EGTA, 10 HEPES, 5 glucose, 4 MgCl₂, 0.5 CaCl₂ (pH 7.3, 270 mOsmol/kg). The pipette to bath resistance ranged from 5 to 7 MOhm. We applied series resistance compensation as much as possible (50–70%). The effective series resistance was in the range of 20–40 MOhm and was checked throughout the whole experiment by using a short depolarizing pulse (10 mV, 20 ms). We accepted recordings only if the series resistance was less than 40 MOhm. Bath solution contained (in mM): 136 NaCl, 2.5 KCl, 20 glucose, 20 HEPES, 2 CaCl₂, 1 MgCl₂ (pH 7.3, 305 mOsmol/kg). We estimated whole-cell input resistance on the basis of passive current responses to moderate depolarizing voltage pulses of short duration (±10 mV for 20 ms). We estimated whole-cell membrane capacitance by integration of the capacitive current transient and division by the respective stimulation voltage. We elicited voltage-gated Na⁺- and K⁺-currents by a series of 200 ms depolarizing pulses applied from the holding potential of −70 mV, in 10 mV increments between −70 and +70 mV. Passive responses were subtracted by using a hyperpolarizing pulse of −20 mV. We recorded spontaneous synaptic currents in voltage-clamp mode at a holding potential of −70 mV without specific blockers. To evaluate action potential generation and discharge properties, we adjusted the cells to −90 mV by steady current injection and depolarized them by injection of positive current pulses (5–50 pA) of 1 s duration under current-clamp conditions. We acquired the signals at a rate of 10 kHz and analyzed using WinTida 5.0 (HEKA Electronics, Lambrecht, Germany).

**_SURF1_ gene augmentation.** We generated lentiviral plasmids expressing SURF1 (amino acid sequence NP_003163.1) and GFP derivative mCitrine using the GatewayTM cloning system[108]. We shuttled open reading frames from entry vectors encoding mCitrine or SURF1 protein (entry clone id: RZPDo839E0486)

into a lentiviral vector harboring a phosphoglycerate kinase (PGK) promoter (pLenti PGK Neo DEST (w531-1). The destination vector was a gift from Eric Campeau & Paul Kaufman (Addgene, Watertown, MA, #19067; http://n2t.net/addgene:19067; RRID: Addgene_19067)[109]. We additionally generated AAV plasmids expressing SURF1 and mCitrine. Therefore, we shuttled the open reading frames of mCitrine and SURF1, respectively, into an AAV destination vector harboring the chicken β-actin promoter (CAG) promoter (pAAV-Gateway). The destination vector was a gift from Matthew Nolan (Addgene, Watertown, MA, #32671; http://n2t.net/addgene:32671; RRID: Addgene #32671)[110]. The preparation of lentiviral and AAV serotype 9 (AAV9) particles was performed by the Viral Core Facility of the Charité Berlin as previously described[111]. For lentiviral transduction, we seeded NPCs onto 6-well plates at a concentration of ~500,000 cells per well. The next day, we transduced the cells with viruses expressing either mCitrine or WT-SURF1. We used a titer of 1.85E + 08 particles per ml for mCitrine and 1.32E + 08 particles per ml for WT-SURF1. One day later, we changed the medium kept the cells for 2 days in the incubator. Then, we carried out antibiotic selection using 500 µg per ml of geneticin (G418; Gibco/Thermo Fisher Scientific, Waltham, MA, USA, #10131027) for 5 days according to the related killing curves. We established a stable line and let them differentiate into 4w DNs for conducting experiments. For AAV9 transduction, we seeded NPCs or 4w DNs on a seahorse or HCA plate at a concentration of ~12,000 cells per well. The next day, we transduced the cells with viruses expressing either mCitrine or WT-SURF1. We used a titer of 1.32E + 12 particles per ml for mCitrine and 6.55E + 11 particles per ml for WT-SURF1. We transduced the cells 2 days prior to assay day and one day later, we changed the medium to maintain the cells in the incubator until the assay day.

**Treatment strategies**. For hypoxia treatment, we exposed NPCs or 8w DNs to 5% oxygen overnight. The day after, we assessed their bioenergetic profiling and HCA neuronal profiling. We used the following drugs on SURF1 NPCs: α-tocotrienol (AT3) (Cayman Chemical, Ann Arbor, MI, USA, #922500), n-acetyl cysteine (NAC) (Sigma-Aldrich, St. Louis, MO, USA, #A7250), ascorbic acid (AA) (Sigma-Aldrich, St. Louis, MO, USA, #A92902), dihydrolipoic acid (DHLA) (Sigma-Aldrich, St. Louis, MO, USA, #T8260), increased glucose, and pyruvate supplementation. We treated SURF1 NPCs ON with bezafibrate (Selleckchem, Houston, TX, USA, #S4159). For lentiviral-mediated PGC1A overexpression, we first obtained the PGC1A open reading from the pcDNA4 myc PGC1A plasmid, which was a gift from Toren Finkel (Addgene, Watertown, MA, #10974; http://n2t.net/addgene:10974; RRID: Addgene_10974). We amplified the PGC1A open reading frame using overhang forward and reverse primers reported in Supplementary Data 12. We introduced a stop codon at the C-terminus and GatewayTM-compatible BP sites flanking the open reading frame. The resulting PCR product was cloned using BP Clonase (Invitrogen, Thermo Fisher Scientific, Waltham, MA, USA) into a pDONR221 entry plasmid, which was subsequently used to shuttle the PGC1A open reading frame into the pLenti PGK Neo DEST vector using LR Clonase (Invitrogen, Thermo Fisher Scientific, Waltham, MA, USA). The resulting plasmid was analyzed by restriction enzyme digest with BsrGI and sequence identity was confirmed by Sanger sequencing. The preparation of lentiviral particles was performed by the Viral Core Facility of the Charité Berlin as previously described[111]. We transduced the viral particles on SURF1 NPCs (SURF1_Mut: S1, S2) following the procedure described for SURF1 gene augmentation.

**Statistics and reproducibility**. We analyzed the data using GraphPad Prism 8.0 (GraphPad Software, Inc.) and employed R environment for statistical computing. For all datasets, we tested the normality of the distribution using GraphPad Prism. Unless otherwise indicated, we expressed the data as mean and standard error of the mean (mean ± s.e.m.). $P$ values below 0.05 were considered significant. We performed outlier test analysis to identify potential outliers using GraphPad Prism (www.graphpad.com/quickcalcs/Grubbs1.cfm). The data are presented as scatter plots with individual data points showing all individual measurements. We carried out all experiments using at least three biological replicates over different independent experiments. In all respective figures legends, we report the exact number of biological replicates and independent experiments. We assessed statistical significance using parametric tests (Student's $t$-test, ANOVA) for normally-distributed data and non-parametric tests (two-sided Mann–Whitney U test, Kruskal–Wallis) when normal distribution could not be verified. The statistical details of the experiments can be found in the respective figure legends.

**Reporting summary**. Further information on research design is available in the Nature Research Reporting Summary linked to this article.

## Data availability
There are restrictions to the availability of the patient-derived iPSC lines generated in this study due to the nature of our ethical approval that does not support sharing to third parties without a specific amendment and does not allow to perform genomic studies to respect the European privacy protection law. The datasets generated during this study are available, in case data protection laws did not prevent the original datasets from being published: Single-cell RNA-sequencing dataset: deposited in Gene Expression Omnibus (GEO) database, accession number: GSE152915. RNA-sequencing dataset: deposited in

GEO database, accession number: GSE126360. Proteomics dataset: deposited in the ProteomeXchange Consortium server, project accession: PXD019112. Metabolomics dataset: deposited in PeptideAtlas, identifier: PASS01598. Source data are provided with this paper.

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

## Acknowledgements

We thank Carlo Viscomi (Cambridge, UK) for critical reading of the initial manuscript. We are grateful to Celina Kassner, Pia Badstuebner, and Luica Delius for technical support with cell culture, Anastasiya Boltengagen for technical support with single-cell RNA-seq, and Beata Lukaszewska-McGreal for proteome and metabolome preparation. We acknowledge support from MDC (BOOST award to A.P. and M.S.), Deutsche Forschungsgemeinschaft (DFG) (PR1527/1-1 & PR1527/5-1 to A.P.; Germany's Excellence Strategy – EXC-2049-390688087 and SCHU1187/4-1 to M.S.), Spark and Berlin Institute of Health (BIH) (BIH Validation Funds to A.P. and M.S.), United Mitochondrial Disease Foundation (UMDF) (Roadmap for Leigh to A.P.), University Hospital Düsseldorf (Forschungskommission UKD to A.P.), National Science Center of Poland (No. 2016/22/M/NZ2/00548 to P.L.), DZNE and UK DRI (Programme Award to J.P.), Charité University Medicine–Universitätsmedizin Berlin (3R-Grant "Replacement of experimental animals by human stem cell-based models for investigation of neurodevelopmental disorders" to M.S.), Biometra PSR2019-Brunetti (to D.B), Austrian Science Fund (FWF) [I 4695-B, GENOMIT] to J.A.M., and the German Federal Ministry of Education and Research (BMBF) (e:Bio young investigator grant AZ.031A318 and 031L0211 to A.P. and AERIAL P1 to J.P.).

## Author contributions

Conceptualization: A.P., G.I., M.S. Methodology: G.I., A.R.-W., P.L., T.M.P., R.J., E.B., D.B., A.Z., C.S., W.S., T.H., U.C., S.D., D.B., M.D., E. B., M.-T.H., M.B.L., D.M., J.P. Formal analysis: T.M.P., B.M., K.U., P.G., N.K. Resources: A.P., E.E.W; W.S.; N.R.; R.K., M.S., M.D.V., J.A.M., M.G., D.P.J., J.A.M., J.P., S.B.W., E.M. Writing—original draft: A.P. Writing—review & editing: A.P., G.I., M.S. Supervision: A.P., R.K., E.E.W., N.R., M.S.; Visualization: G.I., A.P. Funding acquisition: A.P., M.S.

## Funding

## Competing interests

The authors declare no competing interests.
