## [Peer Review File · Nature Communications]

Reviewer #1 (Remarks to the Author):

In this manuscript authors generate a human model of Leigh syndrome caused by mutations in the complex IV assembly gene, SURF1. They show compromised early neuronal morphogenesis in neuronal cultures and cerebral organoids using single-cell RNA-sequencing and multi-omics analysis. The defects emerged at the level of NPCs, which retained a glycolytic proliferative state that failed to instruct neuronal morphogenesis.

This is not the first LS iPSC-based model because other groups, including the authors, have previously generated neuronal models of LS patients harbouring mutations in complex V and complex I mtDNA genes. In this sense, there is a certain lack of novelty. However, manuscript contains data that could be of interest for the journal although several important issues must be addressed before accepting it.

Major changes:

1.General comment: In general, manuscript content is not clearly presented. It is not well structured and it is difficult to understand all the ideas that the authors want to convey to the reader. In some sections is even difficult to know what cell types have been used in each experiment. The experiments must be understood without the necessity of rummaging around all the sections, i.e. when reading the section "SURF1 mutations impair neuronal maturation" it is not clear what iPSC lines have been employed for the neuronal differentiation assay. Only when you read the figure and supplementary figure legends you can extract the information. The same applies for other sections in the manuscript. The manuscript text must be understandable by itself. Otherwise, data can be very hard to follow. Then, I strongly suggest authors to reconsider what is written to clarify it.

2.In the introduction section (lines 89-92) authors state: "Recent studies generated induced pluripotent stem cells (iPSCs) from LS patients carrying mtDNA mutations in Complex V (CV) gene MT-ATP6. CV defects in patient-derived neural cells caused defective bioenergetics (16, 17), decreased protein synthesis (18), impaired mitochondrial calcium homeostasis (17, 19), and abnormal corticogenesis (20). However, the molecular pathophysiology of LS caused by SURF1 mutations or complex I defects remains to be elucidated". This statement is not exactly so. Not all the generated LS iPSC-based models harbour mutations in genes of the mitochondrial complex V. There is, at least, another model (reference 17) that has been generated from a patient carrying a mutation in the MT-ND5 gene, a gene of the mitochondrial complex I. Please, rewrite this paragraph properly. The same applies to the discussion section.

3.In the biochemical validation of the models, authors use several techniques such as blue native experiments and COX activity measurements in iPSC-derived NPCs. Neurons are less glycolytic than NPCs. What happens in neurons?

4.Authors generate iPSCs from two LS patients (S1 and S2) harbouring different mutations in SURF1. They have also used CRISPR/Cas9 to generate isogenic control lines only for one of the iPSC lines (S2; c.769G>A) and they introduce the same mutation in the control line C1. Why only this mutation? They must follow the same procedure for both mutations (not only for c.769G>A).

5.Not all the experiments have been performed using the same cell lines. This way results could be biased.

6.Authors perform several bioenergetics experiments but they don't explain clearly in what culture medium have been carried out. Have they used a medium without glucose and containing galactose as a carbon source? If not, it would be very interesting to promote ATP production by OXPHOS.

Minor changes:

1.Please, include the complete nomenclature of all the mutations (not only DNA but also protein) following the rules for the description of sequence variations approved by HGVS

(<https://www.hgvs.org/content/guidelines>).

2. Please, describe all the catalogue numbers of products in the material and methods section. Include also all the countries. Most of the catalogue numbers are missed and there are products including the country and other without it. Please standardize this throughout the manuscript. Without this information it would be really difficult to reproduce the experiments.

Reviewer #2 (Remarks to the Author):

In this excellent report, Inak and collaborators report mechanistic insight into the underlying pathology of Leigh Syndrome (or sub-acute necrotizing encephalomyelopathy), a rare genetic disease affecting brain development and thought to be caused by defective oxidative metabolism. This study is an extraordinary effort in characterizing the effect of SURF-1 mutations in iPSC-derived systems including neural progenitor cells and brain organoids. This is one of the first reports of systematic characterization of the pathological consequences of SURF1 mutations in human development that is not limited to clinical reports. The authors expand the number of tools available to study Leigh syndrome. Using CRISPR, the authors corrected the SURF-1 mutations generating true isogenic controls and preserving the genetic background of patient cells. They went one step further and introduced the SURF-1 mutations in a WT iPSC line. After confirming that the LS iPSC model recapitulated the disease at the biochemical level, they also provide a complete characterization of the effects of SURF1 mutations, which are among the most frequent causes of Leigh Syndrome, on neurogenesis. Besides generating a model system for LS caused by SURF-1 mutations, the authors also investigated the potential benefits of activating mitochondrial biogenesis in SURF-1 mutant NPCs. Inak and collaborators suggest that increasing mitochondrial biogenesis could alleviate the pathological consequences of SURF-1 mutations. In my opinion, this paper is timely and ready for publication with only minor suggestions to improve content and clarity. No additional experiments are required.

1. Line 78, describe the meaning of SURF1 abbreviation: Surfeit locus protein 1.

2. Describe in more detail the studies that led to the identification of SURF1 as a key regulator of Complex IV assembly.

3. Additional discussion about how SURF-1 mutations could affect other types of neurons of the cortex besides dopaminergic neurons, and also potential explanations or speculation as to why fibroblasts which rely also on oxidative phosphorylation do not have any apparent bioenergetic defect. Are fibroblasts compensating by increasing mitochondrial biogenesis? No experiments are needed, only increased discussion is recommended.

4. It would be nice to know for each figure exactly how many organoids were used and from which differentiation (same or different). There are some overall details in the statistics section, but I didn't see these details for each experiment.

5. The pdf file may have been corrupted and thus, printing the figures caused problems. This could be an issue caused by the size of the file, but it may be important to consider when preparing the final version of the paper.

6. Line 289: "TUJ1-positive neurons within 4w and 8w SURF1 DNs..." This sentence needs to be restructured for clarity.

7. About the format of the references, the numbers should be placed before periods in each sentence.

Reviewer #3 (Remarks to the Author):

Summary

In this multi-authored study, patient-specific pluripotent stem cells and CRISPR/CAS-9 engineering were used to develop a human model of Leigh Syndrome (LS) with mutations in the complex IV gene SURF1. The use of multi-omic approach (transcriptomics, proteomics and metabolomics) combined with additional phenotypic characterization (e.g. bioenergetics, neural morphogenesis) enabled the demonstration that the generated SURF1 defective neural progenitor cells (NPC) and organoids remain in a glycolytic proliferative state and fail to undergo morphogenesis. Interventions augmenting normal SURF1 gene or PGC1a expression, but not hypoxia or antioxidants, restored neuronal morphogenesis.

Critique

General comments

This is a very impressive piece of work. It is a well conducted and original study that provides extremely important and novel information about the molecular mechanisms underlying human neuronal pathology in LS, specifically with SURF1 mutations.

The lack of effective study models in the field of inherited mitochondrial disorders has been and remains a major limitation for the understanding of molecular mechanisms underlying patient phenotypes, particularly those of neurological manifestations, and for the development of effective therapeutic interventions. While commonly used models such as skin fibroblasts from patients have been useful in providing some insights, they have reached their limitations given that these cells do not recapitulate the bioenergetics needs of neural cells. Similarly, mouse models have been quite frustrating in that in many cases they often do not recapitulate the patient phenotype.

Hence, the findings in this study that SURF1 NPCs retain a glycolytic proliferative phenotype that failed to instruct neuronal morphogenesis is novel and extremely important in advancing the field. Even more important is the demonstration that metabolic reprogramming of these cells by augmenting normal SURF1 gene expression or PGC1a expression using bezafibrate (BZ) enable the metabolic shift to a more oxidative phenotype and restore morphogenesis. These results suggest concrete intervention strategies based on a clearly identified mechanism.

Additional findings from these studies are also interesting, which relates to treatments that are currently or commonly put forward for inherited mitochondrial disorders, namely antioxidants and hypoxia, which were found not to be beneficial.

In general, this reviewer was very enthusiastic about this study and had only few specific concerns, both conceptual and methodological, that are listed below, which need to be addressed to strengthened this study.

Specific comments

A) Results and data interpretation

A1) Findings in SURF-1 mutant fibroblasts (Page 12, lines 306-308; Fig. 5g; Suppl. Fig. 5e).

Based on their findings, the authors conclude that there is no bioenergetics defects in SURF1-mutant fibroblasts (Fig. 5g and Suppl. Fig. 5e) and thus, SURF1 mutations might cause a metabolic impairment particularly in neural cells. Intuitively, one would expect that the bioenergetic defects in SURF1-mutant fibroblasts could be less important than in their neural cell counterparts, but not necessarily absent. In fact, while SURF1-mutant fibroblasts appear to have basal OCR similar to controls (Fig. 5g), this does not seem to be the case for the maximal rate, which to this reviewer's understanding of Suppl. Fig. 5e appears to be reduced (even with n=2

independent experiments). The measurements of other bioenergetic read-outs such as ECAR or lactate release, could strengthened this conclusion or else this statement should be modulated.

A2) Recapitulation of findings in the complex 1 gene NDUFS4 cells (Page 13, lines 331-338, Fig. 5q, Fig. 5r-s and Suppl. Fig. 5I).

From findings with iPSC-derived NPCs carrying NDUFS4 mutations, the authors conclude that early neuronal morphogenesis defects might represent a possible general mechanisms of LS. These findings include reduction of neurite outgrowth (Fig. 5r-s) and of mitochondrial membrane potential (MMP; Fig. 5q). The MMP is the only bioenergetic read-out that is reported in these cells, which is one that has not been previously assessed for SURF-1 mutant-derived cells. The measurements of other bioenergetics read-outs in NPCs carrying NDUFS4 mutations, such as basal ECAR, basal OCR, maximal respiration or lactate release, would strengthen the comparison and conclusion. Otherwise, this conclusions appears as an overstatement and should be modulated.

A3) BZ treatment effect on proliferative and pluripotency-associated markers (c-MYC and OCT4).

From data shown in Fig. 7I, the lowering effect of BZ treatment was not immediately apparent to this Reviewer. Please clarify the data presentation, which is different from other panels.

A4) Significance of findings of similar mitochondrial ROS in SURF1 NPCs and CTL at basal level and after mitochondrial stress (Suppl. Fig. 5 i-j) and lack of beneficial effects of antioxidants in SURF1 NPCs.

In inherited mitochondrial disorders such as LS, enhanced oxidative stress and accordingly interventions with antioxidants are commonly put forward by many (albeit not all) investigators. The fact that this notion is not supported by findings in SURF1 NPCs would deserve additional comments in the discussion. This may be of relevance to the general lack of benefits of antioxidant interventions reported in those patients. Similar considerations apply to hypoxia treatment, which was not effective in SURF1 NPCs. This should also be briefly discussed because of relevance to current thinking about treatments in patients with inherited mitochondrial disorders.

A5) Impact of SURF1 defect in mouse versus human neural cells.

In the Introduction, the authors put forward the importance of developing human cell models of LS based on the lack of effective model systems. As an example for SURF1 mutations, it is mentioned that SURF1 KO mice failed to recapitulate the neurological phenotype. Intuitively, one would expect that mouse neuronal cells would also need to undergo a metabolic shift towards oxidative metabolism to undergo morphogenesis. Following their study, do the authors have any clues as to why the mouse model failed to recapitulate the neurological phenotype.

B) Statistics and data presentation/report

B1) According to the methods section, data are shown as mean with SD or SEM following testing for normality.

In most instances in this study, data shown as means with SEM are from 3 or even 2 independent experiments. Please clarify how normality was tested and justify the use of mean with SEM for the majority of figures.

B2) Format of figures vary from one panel to another, event in the same figure.

Please justify or else use the same format for uniformity and to facilitate understanding of results. Results are mostly from 2 or 3 independent experiments, yet there are more than 2 or 3 points for each group.

B3) Page 10, lines 257-258: "Community 2 comprised the majority of metabolites and pathways associated with OXPHOS function (Fig. 4b, Suppl. Fig. 4h, Suppl. Table 4)."

The referenced figures and tables do not appear to clearly support this statement. Please verify and clarify or else modify the sentence to enhance understanding.

B4) Supplemental table 8 – Metabolomic data; Page 42, lines 1133-1135).

According to the text, this table reports a list of all metabolites, including MRM ratios, retention times, KEGG and HMDB identifiers. Please provide the MRM ratios and retention times, which are not included. Please also provide SD on intensity values as well as p-values and q-values. Furthermore, ratios in this table were calculated as Control (S2)/SURF1 (S2_Corr1), which is the reverse of what is shown in Figure 4E, which is misleading. Please correct.

B5) Page 11, lines 266: The text lists p-hydroxybenzoic acid among glycolytic intermediates. Please remove.

B6) Measurement of ATP production. This has not been mentioned or referenced in the method section.

REVIEWER COMMENTS

Reviewer #1 (Remarks to the Author):

In this manuscript authors generate a human model of Leigh syndrome caused by mutations in the complex IV assembly gene, SURF1. They show compromised early neuronal morphogenesis in neuronal cultures and cerebral organoids using single-cell RNA-sequencing and multi-omics analysis. The defects emerged at the level of NPCs, which retained a glycolytic proliferative state that failed to instruct neuronal morphogenesis.

This is not the first LS iPSC-based model because other groups, including the authors, have previously generated neuronal models of LS patients harbouring mutations in complex V and complex I mtDNA genes. In this sense, there is a certain lack of novelty. However, manuscript contains data that could be of interest for the journal although several important issues must be addressed before accepting it.

Answer:

The reviewer is correct that it is not the first report of iPSCs carrying mutations associated with Leigh syndrome (LS). However, to our knowledge this is the first work reporting iPSCs carrying *SURF1* mutations, which are among the most common single-gene defects causing LS. We also confirmed key deficiencies of early neuronal morphogenesis in neural cells carrying mutations in the gene *NDUFS4*. *NDUFS4* mutations are another frequent cause of LS, and have also not been studied so far using patient-specific iPSCs. Therefore, we think that our work is indeed novel, as it reports unpublished models of LS.

We would also like to point out that the novelty and relevance of our manuscript does not lie only in the fact that we model LS with yet unpublished iPSC lines, but also that we unveil a previously unknown pathogenetic mechanism of this incurable disease that might be targeted for medical intervention.

So far LS was considered an early-onset neurodegenerative disease where the neurons develop normally and then start degenerating because of free radical damage. Our study demonstrates that there is an additional neurodevelopmental component to this disease. We found that mutant NPCs fail to undergo a proper metabolic programming and therefore impair physiological neurogenesis and neuronal morphogenesis. These findings are in agreement with the occurrence of microcephaly and neurodevelopmental defects in children with LS. We also failed to observe excessive ROS production in *SURF1* neural cells or phenotypic ameliorations following antioxidants. These results may help explaining the lack of clinical success for antioxidant treatments in LS. Hence, our work might lead to a shift in the perspective of LS pathogenesis and can open the way to the development of novel therapeutic approaches targeting NPCs and their metabolic programming towards OXPHOS.

In order to better convey this message and highlight the novelty of our work, we updated and modified the text in the introduction and discussion.

Major changes:

1. General comment: In general, manuscript content is not clearly presented. It is not well structured and it is difficult to understand all the ideas that the authors want to convey to the reader. In some sections is even difficult to know what cell types have been used in each experiment. The experiments must be understood without the necessity of rummaging around all the sections, i.e. when reading the section “SURF1 mutations impair neuronal maturation” it is not clear what iPSC lines have been employed for the neuronal differentiation assay. Only when you read the figure and supplementary figure legends you can extract the information. The same applies for other sections in the manuscript. The manuscript text must be understandable by

itself. Otherwise, data can be very hard to follow. Then, I strongly suggest authors to reconsider what is written to clarify it.

Answer:

We thank the reviewer for pointing this out. We extensively modified the text and reorganized the data presentation in order to clarify our messages.

We now uniformly present the data as scatter plots in all the figures to clearly show the distribution of all underlying data. We divided the samples into four groups: (i) CTL_NoMut: control-derived iPSCs, (ii) SURF1_NoMut: isogenic controls obtained by biallelic genomic correction of the pathogenic SURF1 mutation in patient-derived iPSCs, (iii) SURF1_Mut: patient-derived iPSCs, (iv) CTL_Mut: isogenic mutants obtained by biallelic genomic introduction of the pathogenic SURF1 mutation in control-derived iPSCs (Fig. 1b-d).

We believe that this data presentation is more transparent and allows the readers to better appreciate the mutation-specific effect and the lack of influence of the genomic background. It is now more evident that corrected iPSC lines behave similarly to control iPSC lines, and that mutated iPSC lines behave similarly to patient iPSC lines.

We now report in the main body of text all the details of the samples used in all experiments. We updated the figure legends to include all the information regarding the lines utilized for each experiment.

2. In the introduction section (lines 89-92) authors state: "Recent studies generated induced pluripotent stem cells (iPSCs) from LS patients carrying mtDNA mutations in Complex V (CV) gene MT-ATP6. CV defects in patient-derived neural cells caused defective bioenergetics (16, 17), decreased protein synthesis (18), impaired mitochondrial calcium homeostasis (17, 19), and abnormal corticogenesis (20). However, the molecular pathophysiology of LS caused by SURF1 mutations or complex I defects remains to be elucidated". This statement is not exactly so. Not all the generated LS iPSC-based models harbour mutations in genes of the mitochondrial complex V. There is, at least, another model (reference 17) that has been generated from a patient carrying a mutation in the MT-ND5 gene, a gene of the mitochondrial complex I. Please, rewrite this paragraph properly. The same applies to the discussion section.

Answer:

The reviewer is correct and we apologize for the lack of clarity. We updated and modified the text in the introduction and discussion sections to include the data from the mentioned study (now reference 29).

3. In the biochemical validation of the models, authors use several techniques such as blue native experiments and COX activity measurements in iPSC-derived NPCs. Neurons are less glycolytic than NPCs. What happens in neurons?

Answer:

It is known since the first description of LS by Danis Leigh in 1951 that LS causes neuronal pathology. For this reason, we performed extensive analyses to elucidate what happens in LS neurons carrying *SURF1* mutations. We found that the OXPHOS pathway is dysregulated in *SURF1* neurons with an increase of glycolysis-associated genes and proteins (see transcriptomics and proteomics in Fig. 4c-d, Supp. Fig. 4h, and Supp. Tables 5-7). Accordingly, *SURF1* neurons exhibited metabolic alterations, including lower OCR and ATP production (Fig. 2c-d), and lower energy-related metabolites like ATP, AMP, ADP (see metabolomics Fig. 4e and Supp. Table 8). These findings demonstrate that *SURF1* mutations cause neuronal bioenergetic defects.

Although we did not measure COX activity in neurons, the same biochemical COX defects might very well likely be present in neurons as well as in NPCs. In fact, even somatic fibroblasts and

muscle tissue that were analyzed during the patient diagnostic work-up showed isolated COX deficiency (see “Subject details” in the methods section). It is thus important to clearly point out that we do not claim that the biochemical COX defects caused by *SURF1* mutations occur only in NPCs. On the contrary, despite a similar underlying biochemical defect, distinct cell types might develop different way of responding and compensating. In fact, it is known that *SURF1* exhibits species-specific and tissue-specific effects on COX assembly (see references 26, 27). What we discovered is that NPCs respond to this biochemical challenge by shifting their metabolism towards glycolysis, thereby hampering early morphogenesis (Fig. 5d-r). This is we think the main message of our work. iPSC-derived NPCs represent the first cell type during neurogenesis to start relying on OXPHOS (as we and others have previously shown, see references 31, 62, 63). Consequently, we found that significant bioenergetic alterations emerge already at the level of NPCs (Fig.5d-h). NPCs carrying *SURF1* mutations fail to shift from glycolysis to OXPHOS, and therefore keep proliferating (Fig. 5n) and undergo improper neuronal morphogenesis (Fig. 5p-r). Consequently, we suggest that, in order to tackle the neuronal pathology of LS, therapeutic strategies might be potentially more effective by targeting the metabolic programming of NPCs (Supp. Fig. 8), since improving NPC function may support the downstream generation of functional neurons (Fig. 7a-f, Fig. 7k-u). We modified the introduction, results, and discussion sections to clarify these points. We also included additional details in the presentation of *SURF1* patients (“Subject details” in the methods section).

4. Authors generate iPSCs from two LS patients (S1 and S2) harbouring different mutations in *SURF1*. They have also used CRISPR/Cas9 to generate isogenic control lines only for one of the iPSC lines (S2; c.769G>A) and they introduce the same mutation in the control line C1. Why only this mutation? They must follow the same procedure for both mutations (not only for c.769G>A).

Answer:

We thank the reviewer for pointing this out. We realized that we did not effectively clarify our genome editing strategy.

We chose to focus on c.769G>A mutation since it has been reported as the most common mutation in the Turkish patient community to which both patients belonged (see reference 35). Moreover, the c.769G>A mutation was clinically more severe (see “Subject details” in the methods section). Consistently, in our *in vitro* experiments S2 cells showed stronger defects compared to S1 cells.

After correcting the c.769G>A mutation in patient S2 iPSCs, we could have corrected also the c.530T>G mutation in patient S1 iPSCs. However, we thought that these experiments would not bring essential new information. In fact, correcting the milder mutation c.530T>G would have been *a priori* more likely to generate isogenic control lines behaving like healthy control lines. Being able to revert the biochemical and cellular defects in the more severely affected cell line would, as we thought, be more convincing. Therefore, we decided to further concentrate our efforts on the more severe c.769G>A mutation. We then introduced this mutation in a control healthy genomic background. We reasoned that introducing the c.769G>A mutation would have a much higher impact, as it would show whether the mutation is sufficient to cause the *in vitro* phenotypes even in a healthy genetic background. This aspect is particularly important for mitochondrial diseases like LS, where the interplay between nuclear and mitochondrial DNA may play a contributing role in the disease pathogenesis (see reference 38). Our genome editing design (summarized in Fig. 1c-d) enabled us to dissect the specific impact of this *SURF1* mutation, regardless of any additional nuclear or mitochondrial genomic differences.

We hope that the reviewer can now understand our genome editing strategy and would agree that performing now the same genome engineering experiments also for the milder c.530T>G

mutation would be extremely labor and cost-intensive, would not bring any new information, and would not change the current conclusions of our study.

In order to clarify our genome engineering strategy and our choice of editing target, we included more details and explanations in the introduction and results sections. We also changed the presentation of Fig. 1, Supp. Fig. 1, and Supp. Fig. 2. Lastly, we included more details in the patient description section to better describe the disease course and clinical severity for the two patients (“Subject details” in the methods section).

5. Not all the experiments have been performed using the same cell lines. This way results could be biased.

Answer:

As mentioned above, the introduction of the c.769G>A mutation into the healthy background was performed at a later stage. Therefore, we have not been able to include these new iPSCs (CTL_Mut) in all the experiments. In order to make this aspect more transparent, we now present all the data with scatter plots divided into four groups: controls (CTL_NoMut), isogenic controls (SURF1_NoMut), patients (SURF1_Mut), and isogenic mutants (CTL_Mut) (Fig. 1d). We believe that this new data presentation clearly indicates the distribution of all underlying data, and thus allows the readers to appreciate which iPSC lines were used for each individual experiment. It is also important to mention that we did not exclude any datasets or any experiments that did not fit our storyline. Hence, we do not think that the data can be considered biased. We have included all the datasets that were prepared and measured, and we are fully transparent with respect to the samples analyzed in each dataset. We also provide all raw data, including raw values of the plots of all main and supplementary figures and uncropped versions of all blots (see Source Data table). We believe that the readers can extract a conclusion based on the presented data that is not biased by our interpretation.

6. Authors perform several bioenergetics experiments but they don't explain clearly in what culture medium have been carried out. Have they used a medium without glucose and containing galactose as a carbon source? If not, it would be very interesting to promote ATP production by OXPHOS.

Answer:

We thank the reviewer for pointing this out. We performed all the experiments (including metabolomics and bioenergetic profiling) using cells grown in glucose-containing medium. We updated the methods section to clarify this important aspect.

It would be indeed interesting to test the effect of metabolic manipulations to promote OXPHOS in LS neural cells. However, glucose-free galactose medium can be detrimental for cells carrying mitochondrial mutations, as it prevents the cells from employing the glycolytic pathway (Robinson, Petrova-Benedict, Buncic, & Wallace. *Nonviability of cells with oxidative defects in galactose medium: a screening test for affected patient fibroblasts*. *Biochem. Med. Metab. Biol.* 48, 122–126, 1992). The negative effect of glucose-free galactose medium was confirmed in cells carrying LS mutations in CI genes (Iannetti, Smeitink, Willems, Beyrath & Koopman. *Rescue from galactose-induced death of Leigh Syndrome patient cells by pyruvate and NAD+*. *Cell Death & Disease* 1135, vol9, 2018). We also previously observed the lack of rescue effects for glucose-free galactose medium in LS-NPCs carrying *MT-ATP6* mutations (see reference 31). For all these reasons, we did not attempt to grow SURF1 NPCs in a medium without glucose and containing galactose as a carbon source.

Nonetheless, we sought to address whether other metabolic manipulations may be able to support OXPHOS and early morphogenesis in SURF1 NPCs. We exposed SURF1 NPCs to increasing amounts of glucose or pyruvate supplementation, which has been suggested as a potential therapy for LS caused by COX defects (see reference 55). These conditions partially

reduced the glycolytic profile of NPCs (Supp. Fig. 7x). However, they were not sufficient to increase OXPHOS (Supp. Fig. 7w) or to improve neurite outgrowth (Supp. Fig. 7y). We included this new set of data and discussed these new findings in the discussion section.

Minor changes:

1. Please, include the complete nomenclature of all the mutations (not only DNA but also protein) following the rules for the description of sequence variations approved by HGVS (<https://www.hgvs.org/content/guidelines>).

Answer:

Thanks for noticing this. We have updated the text.

2. Please, describe all the catalogue numbers of products in the material and methods section. Include also all the countries. Most of the catalogue numbers are missed and there are products including the country and other without it. Please standardize this throughout the manuscript. Without this information it would be really difficult to reproduce the experiments.

Answer:

We have included the missing information.

Reviewer #2 (Remarks to the Author):

In this excellent report, Inak and collaborators report mechanistic insight into the underlying pathology of Leigh Syndrome (or sub-acute necrotizing encephalomyelopathy), a rare genetic disease affecting brain development and thought to be caused by defective oxidative metabolism. This study is an extraordinary effort in characterizing the effect of SURF-1 mutations in iPSC-derived systems including neural progenitor cells and brain organoids. This is one of the first reports of systematic characterization of the pathological consequences of SURF1 mutations in human development that is not limited to clinical reports. The authors expand the number of tools available to study Leigh syndrome. Using CRISPR, the authors corrected the SURF-1 mutations generating true isogenic controls and preserving the genetic background of patient cells. They went one step further and introduced the SURF-1 mutations in a WT iPSC line. After confirming that the LS iPSC model recapitulated the disease at the biochemical level, they also provide a complete characterization of the effects of SURF1 mutations, which are among the most frequent causes of Leigh Syndrome, on neurogenesis. Besides generating a model system for LS caused by SURF-1 mutations, the authors also investigated the potential benefits of activating mitochondrial biogenesis in SURF-1 mutant NPCs. Inak and collaborators suggest that increasing mitochondrial biogenesis could alleviate the pathological consequences of SURF-1 mutations. In my opinion, this paper is timely and ready for publication with only minor suggestions to improve content and clarity. No additional experiments are required.

Answer:

We thank the reviewer for a positive evaluation of our work.

1. Line 78, describe the meaning of SURF1 abbreviation: Surfeit locus protein 1.

Answer:

Thanks for pointing this out. We have included the meaning of the abbreviation.

2. Describe in more detail the studies that led to the identification of SURF1 as a key regulator of Complex IV assembly.

Answer:

The reviewer is correct that the information regarding the role of SURF1 for COX assembly was not clearly presented. We have updated the text in the introduction section to better clarify this important aspect.

3. Additional discussion about how SURF-1 mutations could affect other types of neurons of the cortex besides dopaminergic neurons, and also potential explanations or speculation as to why fibroblasts which rely also on oxidative phosphorylation do not have any apparent bioenergetic defect. Are fibroblasts compensating by increasing mitochondrial biogenesis? No experiments are needed, only increased discussion is recommended.

Answer:

We thank the reviewer for raising these important points. We included explanations/speculations in the discussion section with respect to the potential relevance of LS defects on different neuronal subtypes.

Since also Reviewer 3 raised questions related to the bioenergetics of fibroblasts, we included additional experimental data showing the extracellular acidification rate and lactate release quantification for SURF1 fibroblasts compared to control fibroblasts. These results confirmed the lack of increased glycolysis and lactate release in SURF1 fibroblasts (Fig. 5j-k). Hence, our interpretation is that fibroblasts also experienced COX defects (see “Subject details” in the methods section), but the consequences of such defects are less dramatic than for cells of the neuronal lineage. This phenomenon may be related to the known tissue-specific effect of SURF1 on COX assembly (see references 26, 27) and to the specific importance of COX for neuronal function (see references 64, 65).

We included these new datasets and added additional explanations and speculations in the discussion section. We also included more details in the patient description (“Subject details” in the methods section).

4. It would be nice to know for each figure exactly how many organoids were used and from which differentiation (same or different). There are some overall details in the statistics section, but I didn't see these details for each experiment.

Answer:

We included the details regarding the number of organoids used for each experiment in all respective figure legends.

5. The pdf file may have been corrupted and thus, printing the figures caused problems. This could be an issue caused by the size of the file, but it may be important to consider when preparing the final version of the paper.

Answer:

Thanks for informing us on this. We reduced the size of the figures to avoid problems during the PDF assembly.

6. Line 289: “TUJ1-positive neurons within 4w and 8w SURF1 DNs...” This sentence needs to be restructured for clarity.

Answer:

We restructured this paragraph to clarify the message.

7. About the format of the references, the numbers should be placed before periods in each sentence.

Answer:

Thanks for noticing this. We corrected the typos.

Reviewer #3 (Remarks to the Author):

Summary

In this multi-authored study, patient-specific pluripotent stem cells and CRISPR/CAS-9 engineering were used to develop a human model of Leigh Syndrome (LS) with mutations in the complex IV gene SURF1. The use of multi-omic approach (transcriptomics, proteomics and metabolomics) combined with additional phenotypic characterization (e.g. bioenergetics, neural morphogenesis) enabled the demonstration that the generated SURF1 defective neural progenitor cells (NPC) and organoids remain in a glycolytic proliferative state and fail to undergo morphogenesis. Interventions augmenting normal SURF1 gene or PGC1a expression, but not hypoxia or antioxidants, restored neuronal morphogenesis.

Critique

General comments

This is a very impressive piece of work. It is a well conducted and original study that provides extremely important and novel information about the molecular mechanisms underlying human neuronal pathology in LS, specifically with SURF1 mutations.

The lack of effective study models in the field of inherited mitochondrial disorders has been and remains a major limitation for the understanding of molecular mechanisms underlying patient phenotypes, particularly those of neurological manifestations, and for the development of effective therapeutic interventions. While commonly used models such as skin fibroblasts from patients have been useful in providing some insights, they have reached their limitations given that these cells do not recapitulate the bioenergetics needs of neural cells. Similarly, mouse models have been quite frustrating in that in many cases they often do not recapitulate the patient phenotype.

Hence, the findings in this study that SURF1 NPCs retain a glycolytic proliferative phenotype that failed to instruct neuronal morphogenesis is novel and extremely important in advancing the field. Even more important is the demonstration that metabolic reprogramming of these cells by augmenting normal SURF1 gene expression or PGC1a expression using bezafibrate (BZ) enable the metabolic shift to a more oxidative phenotype and restore morphogenesis. These results suggest concrete intervention strategies based on a clearly identified mechanism.

Additional findings from these studies are also interesting, which relates to treatments that are currently or commonly put forward for inherited mitochondrial disorders, namely antioxidants and hypoxia, which were found not to be beneficial.

In general, this reviewer was very enthusiastic about this study and had only few specific concerns, both conceptual and methodological, that are listed below, which need to be

addressed to strengthened this study.

Answer:

We thank the reviewer for the positive evaluation of our work.

Specific comments

A) Results and data interpretation

A1) Findings in SURF-1 mutant fibroblasts (Page 12, lines 306-308; Fig. 5g; Suppl. Fig. 5e).

Based on their findings, the authors conclude that there is no bioenergetics defects in SURF1-mutant fibroblasts (Fig. 5g and Suppl. Fig. 5e) and thus, SURF1 mutations might cause a metabolic impairment particularly in neural cells. Intuitively, one would expect that the bioenergetic defects in SURF1-mutant fibroblasts could be less important than in their neural cell counterparts, but not necessarily absent. In fact, while SURF1-mutant fibroblasts appear to have basal OCR similar to controls (Fig. 5g), this does not seem to be the case for the maximal rate, which to this reviewer's understanding of Suppl. Fig. 5e appears to be reduced (even with n=2 independent experiments). The measurements of other bioenergetic read-outs such as ECAR or lactate release, could strengthened this conclusion or else this statement should be modulated.

Answer:

The reviewer is correct that SURF1 fibroblasts are not entirely unaffected. In fact, skin fibroblasts were used during the diagnostic work-up of the patients and showed defective COX activity (see "Subject details" in the methods section).

In order to clarify this important point, we included ECAR and lactate release measurements of SURF1 fibroblasts. SURF1 fibroblasts did not show increased glycolytic profile compared to control fibroblasts. Lactate release levels were not changed (Fig. 5k) and ECAR levels were even lower in SURF1 fibroblasts than in control fibroblasts (Fig. 5j). We also included ECAR data for iPSCs that also showed no upregulation (Supp. Fig. 5h).

Overall, our findings indicated that (at least in our hands under our experimental conditions) neural cells (NPCs, neurons, and organoids) showed an OXPHOS impairment and glycolytic shift that was not as evident as in the case of fibroblasts or iPSCs.

We included these new datasets and updated and modified the results and discussion sections. We now more clearly state that patient fibroblasts presented biochemical or bioenergetic defects, but that these defects were not sufficient to trigger extensive OXPHOS impairment or an upregulation of the glycolytic pathway, as we observed in the case of NPCs.

A2) Recapitulation of findings in the complex 1 gene NDUFS4 cells (Page 13, lines 331-338, Fig. 5q, Fig. 5r-s and Suppl. Fig. 5l).

From findings with iPSC-derived NPCs carrying NDUFS4 mutations, the authors conclude that early neuronal morphogenesis defects might represent a possible general mechanisms of LS. These findings include reduction of neurite outgrowth (Fig. 5r-s) and of mitochondrial membrane potential (MMP; Fig. 5q). The MMP is the only bioenergetic read-out that is reported in these cells, which is one that has not been previously assessed for SURF-1 mutant-derived cells. The measurements of other bioenergetics read-outs in NPCs carrying NDUFS4 mutations, such as basal ECAR, basal OCR, maximal respiration or lactate release, would strengthen the comparison and conclusion. Otherwise, this conclusions appears as an overstatement and should be modulated.

Answer:

The reviewer is correct that we have not performed in-depth analysis of cellular bioenergetics for *NDUFS4*-mutant NPCs. We modified the text in the abstract, results, and discussion to tone down all our claims related to *NDUFS4* mutations.

A3) BZ treatment effect on proliferative and pluripotency-associated markers (c-MYC and OCT4).

From data shown in Fig. 7l, the lowering effect of BZ treatment was not immediately apparent to this Reviewer. Please clarify the data presentation, which is different from other panels.

Answer:

We reorganized the data presentation throughout all figures. We now have a more uniformed visual representation based on scatter plots showing all underlying individual data points. The graph mentioned by the reviewer has also been updated. We believe that the effect of BZ treatment on the expression of MYC and OCT4 in SURF1 NPCs is now more clearly evident (Fig. 7l-m).

A4) Significance of findings of similar mitochondrial ROS in SURF1 NPCs and CTL at basal level and after mitochondrial stress (Suppl. Fig. 5 i-j) and lack of beneficial effects of antioxidants in SURF1 NPCs.

In inherited mitochondrial disorders such as LS, enhanced oxidative stress and accordingly interventions with antioxidants are commonly put forward by many (albeit not all) investigators. The fact that this notion is not supported by findings in SURF1 NPCs would deserve additional comments in the discussion. This may be of relevance to the general lack of benefits of antioxidant interventions reported in those patients. Similar considerations apply to hypoxia treatment, which was not effective in SURF1 NPCs. This should also be briefly discussed because of relevance to current thinking about treatments in patients with inherited mitochondrial disorders.

Answer:

We thank the reviewer for suggesting this. We included a section in the discussion to comment on the implications of our antioxidants-related findings and to speculate about future effective intervention strategies for LS.

A5) Impact of SURF1 defect in mouse versus human neural cells.

In the Introduction, the authors put forward the importance of developing human cell models of LS based on the lack of effective model systems. As an example for SURF1 mutations, it is mentioned that SURF1 KO mice failed to recapitulate the neurological phenotype. Intuitively, one would expect that mouse neuronal cells would also need to undergo a metabolic shift towards oxidative metabolism to undergo morphogenesis. Following their study, do the authors have any clues as to why the mouse model failed to recapitulate the neurological phenotype.

Answer:

In order to better clarify this important point, as also mentioned in our response to Reviewer 2, we modified the introduction to include the description of previous studies conducted on human and mouse fibroblasts that demonstrated that human cells appear more dependent on SURF1 for COX assembly than mouse cells (see references 26, 27). These species-specific differences in COX assembly might potentially be at the bases for the lack of phenotypes in *Surf1*-KO mice. We also updated the discussion section to include these considerations.

B) Statistics and data presentation/report

B1) According to the methods section, data are shown as mean with SD or SEM following testing for normality.

In most instances in this study, data shown as means with SEM are from 3 or even 2 independent experiments. Please clarify how normality was tested and justify the use of mean with SEM for the majority of figures.

Answer:

We performed normality tests using GraphPad Prism. For easier data visualization, we now uniformly use SEM (with the exception of electrophysiology curves where we use SD). We are fully transparent with our data representation, since we now show all the individual data points of all experiments with scatter plot graphs for all the figures. We updated the methods section to clarify these aspects.

B2) Format of figures vary from one panel to another, event in the same figure.

Please justify or else use the same format for uniformity and to facilitate understanding of results. Results are mostly from 2 or 3 independent experiments, yet there are more than 2 or 3 points for each group.

Answer:

We thank the reviewer for raising these important points. As mentioned in our response to Reviewer 1, we have reorganized and restructured our data representation. We now show all the samples divided into four groups: controls (CTL_NoMut), isogenic controls (SURF1_NoMut), patients (SURF1_Mut), and isogenic mutants (CTL_Mut) (see Fig. 1d). We believe that this new data visualization is clearer and allows the readers to more easily to appreciate the effect of *SURF1* mutations regardless of the genomic background. We also homogenized the data visualization. Throughout whole figures, we now uniformly use the same graph style (scatter plots showing all individual data points) that clearly show the distribution of all underlying data. We apologize for not being clear regarding the source of the individual data points. Several experiments (such as Seahorse profiling or HCA-based quantification of branching) are conducted on multi-well format where different cells are plated and measured. We then repeated the same measurements in various independent experiments. Therefore, at the end there are several data points representing the total of individual measurements obtained in all independent experiments. In order to be more transparent, we now present all the individual measurements as individual data points. We updated the statistical methods section and respective figure legends to clarify these issues.

B3) Page 10, lines 257-258: "Community 2 comprised the majority of metabolites and pathways associated with OXPHOS function (Fig. 4b, Suppl. Fig. 4h, Suppl. Table 4)."

The referenced figures and tables do not appear to clearly support this statement. Please verify and clarify or else modify the sentence to enhance understanding.

Answer:

We rewrote this paragraph to clarify the message.

B4) Supplemental table 8 – Metabolomic data; Page 42, lines 1133-1135).

According to the text, this table reports a list of all metabolites, including MRM ratios, retention times, KEGG and HMDB identifiers. Please provide the MRM ratios and retention times, which are not included. Please also provide SD on intensity values as well as p-values and q-values. Furthermore, ratios in this table were calculated as Control (S2)/SURF1 (S2_Corr1), which is the reverse of what is shown in Figure 4E, which is misleading. Please correct.

Answer:

We thank the reviewer for detecting this mistake. We now provide the corrected Supplementary Table 8 including all the requested information. We also updated the respective methods section.

B5) Page 11, lines 266: The text lists p-hydroxybenzoic acid among glycolytic intermediates. Please remove.

Answer:

Thanks for noticing this. We removed the mentioned text.

B6) Measurement of ATP production. This has not been mentioned or referenced in the method section.

Answer:

Thanks for noticing this. We updated the methods section.

Reviewer #1 (Remarks to the Author):

Authors have improved very much the manuscript content. They have responded properly all the issues raised by this reviewer.

Reviewer #2 (Remarks to the Author):

The authors successfully addressed all my comments and suggestions. In my opinion, this article is ready for publication.

Reviewer #3 (Remarks to the Author):

The authors should be commended for their thorough response to criticisms, including those from this reviewer. The revised manuscript is considerably strengthened both regarding conceptual and methodological aspects. This made this reviewer even more enthusiastic about this study, although a few minor remaining issues still need to be addressed.

1) Metabolomic analysis.

Please report the cut-off p-value and q-value that was applied on metabolomic data either in Figure 4 Legend or in the Method section.

2) Page 12 - line 311: G6PD1

G6PD1 is a cytosolic enzyme part of the oxidative pentose phosphate pathway. It is not directly involved in mitochondrial bioenergetics. Please correct.

3) Lack of impact of antioxidant treatment

The authors might like to cite the work of others that have also reported results that do not support the use of antioxidants to treat LS, including N-acetylcysteine in fibroblast cells derived from patients with Leigh Syndrome French Canadian type, also resulting in a tissue-specific COX deficiency (PMID: 25835550; Burelle et al. PLoS One 2015).

Point-by-point response to reviewer comments

Reviewer #1 (Remarks to the Author):

Authors have improved very much the manuscript content. They have responded properly all the issues raised by this reviewer.

Thank you for the positive evaluation.

Reviewer #2 (Remarks to the Author):

The authors successfully addressed all my comments and suggestions. In my opinion, this article is ready for publication.

Thank you the positive evaluation.

Reviewer #3 (Remarks to the Author):

The authors should be commended for their thorough response to criticisms, including those from this reviewer. The revised manuscript is considerably strengthened both regarding conceptual and methodological aspects. This made this reviewer even more enthusiastic about this study, although a few minor remaining issues still need to be addressed.

1) Metabolomic analysis.

Please report the cut-off p-value and q-value that was applied on metabolomic data either in Figure 4 Legend or in the Method section.

Thank you for noticing the missing information regarding the statistical analysis of metabolomics. We included the missing text in the methods section and slightly modified the Fig. 4e legend.

2) Page 12 - line 311: G6PD1

G6PD1 is a cytosolic enzyme part of the oxidative pentose phosphate pathway. It is not directly involved in mitochondrial bioenergetics. Please correct.

Thank you for noticing this mistake. We removed G6PD1 from the list of genes involved mitochondrial bioenergetics and rewrote the section related to NAD and NADP metabolism.

3) Lack of impact of antioxidant treatment

The authors might like to cite the work of others that have also reported results that do not support the use of antioxidants to treat LS, including N-acetylcysteine in fibroblast cells derived from patients with Leigh Syndrome French Canadian type, also resulting in a tissue-specific COX deficiency (PMID: 25835550; Burelle et al. PLoS One 2015).

We included the mentioned reference and discussed it in the discussion section.